# LLMs Process Lists With General Filter Heads

**Arnab Sen Sharma**[*]**, Giordano Rogers, Natalie Shapira, and David Bau**
Khoury College of Computer Sciences, Northeastern University

## ABSTRACT

We investigate the mechanisms underlying a range of list-processing tasks in LLMs, and we find that LLMs have learned to encode a compact, causal representation of a general filtering operation that mirrors the generic "filter" function of functional programming. Using causal mediation analysis on a diverse set of list-processing tasks, we find that a small number of attention heads, which we dub *filter heads*, encode a compact representation of the filtering predicate in their query states at certain tokens. We demonstrate that this predicate representation is general and portable: it can be extracted and reapplied to execute the same filtering operation on different collections, presented in different formats, languages, or even in tasks. However, we also identify situations where transformer LMs can exploit a different strategy for filtering: eagerly evaluating if an item satisfies the predicate and storing this intermediate result as a flag directly in the item representations. Our results reveal that transformer LMs can develop human-interpretable implementations of abstract computational operations that generalize in ways that are surprisingly similar to strategies used in traditional functional programming patterns.

## 1 INTRODUCTION

When asked to *find the fruit* in a list, language models reveal a surprisingly systematic mechanism: they don't solve each filtering task anew, but instead encode predicates into portable representations. This neural representation of *"is this a fruit?"* can be extracted from one context and applied to a different list, presented in a different format, in a different language, and to some extent to a different task. These abstract, reusable operations suggest that transformers develop modular computational primitives rather than task-specific heuristics.

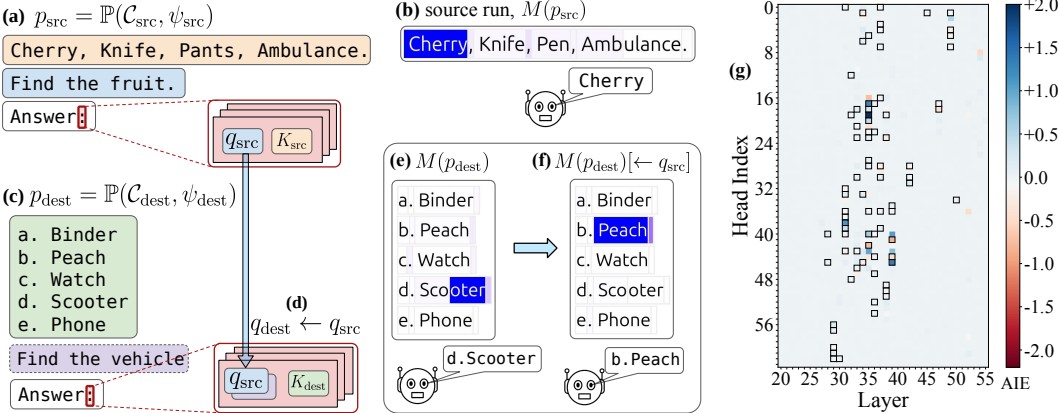

Figure 1: A filter head [35, 19] in Llama-70B encodes a compact representation of the predicate *"is this fruit?"*. **(a)** Within a prompt $p_{\text{src}}$ to find a fruit in a list, we examine the attention head's behavior at the last token ":" **(b)** The head focuses its attention on the one fruit in the list. **(c)** We examine the same attention head's behavior in a second prompt $p_{\text{dest}}$ searching a different list for a vehicle **(d)** and we also examine the behavior of the head when patching its query state to use the $q_{\text{src}}$ vector from the source context. **(e)** The head attends to the vehicle but then **(f)** redirects its attention to the fruit in the new list after the query vector is patched. **(g)** A sparse set of attention heads work together to conduct filtering over a wide range of predicates. These filter heads are concentrated in the middle layers (out of 80 layers in Llama-70B).

[*]Correspondence to sensharma.a@northeastern.edu. Website `filter.baulab.info`.

To understand this phenomenon systematically, we turn to Marr's three levels of analysis (Marr, 1982). At the *computational* level, we identify what is being computed: the selection of elements satisfying a predicate. At the *algorithmic* level, we reveal how this is achieved: through a three phase computation corresponding to a *map*, *filter*, and *reduce*, occurring in that order. The *map* step is equivalent to populating the latents of the items in a list with the right associations or semantic information, a step that has been documented in prior literature (Geva et al., 2023; Meng et al., 2022). In this work we focus on the non-trivial computation step, *filter*, that follows after map. At the *implementation* level, we reveal how filtering is implemented in LMs: through specialized attention heads, which we dub *filter heads*, that encode predicates as geometric directions in query space. We find that these heads, concentrated in the middle layers of the LM, remain largely shared even as the specific predicate varies. This framework allows us to move beyond simply observing that LMs can filter, to understanding the explicit mechanisms through which list-processing operations emerge from the transformer architecture.

Our analysis yields three key insights:

**Localized Mechanism.** The list processing algorithm is implemented in a consistent set of localized components: a set of attention heads that we call filter heads. These heads encode a "compiled" representation of the predicate as query states at specific tokens — typically where the LM is required to produce its answer. These query states interact with the key states that carry semantic information of the list items, producing attention patterns that select the items satisfying the predicate.

**Generalization.** These filter heads are not specific to a single predicate, but can encode a distribution of predicates. And this encoding is sufficiently abstract that it can be extracted from one context and transported to another context to trigger the same filtering operation on a different collection of items, presented in a different format, in a different language, even in a different *reduce* task that follows after the filtering step.

**Computational Redundancy.** Additionally, our investigations reveal that LMs can perform filtering in two complementary ways: lazy evaluation via filter heads vs eager evaluation by storing *is_match* flags directly in the item latents. This dual implementation strategy mirrors the fundamental lazy/eager evaluation strategies in functional programming (Henderson & Morris Jr, 1976; Friedman et al., 1976). This second route reveals a broader principle in neural computations: transformer LMs can maintain multiple pathways for the same operation (McGrath et al., 2023; Wang et al., 2023) and can dynamically select between them based on what information is available.

We validate these findings through experiments across six different filter-reduce tasks of varying complexity, each requiring the LM to filter based on different information before performing a reduce step to provide a specific answer. We test the portability of the "compiled" predicate across different presentation format, language, and tasks. We conduct ablation studies to confirm the necessity of filter heads when the LM performs filter operations. Finally, we demonstrate that the learned predicate representations can serve as zero-shot probes for concept detection, offering a training-free alternative to traditional linear probing methods.

## 2 METHOD

### 2.1 BACKGROUNDS AND NOTATIONS

**Language Model.** An autoregressive transformer language model, $M : \mathcal{X} \to \mathcal{Y}$ over a vocabulary $\mathcal{V}$, maps a sequence of tokens $x = \{x_1, x_2, \ldots, x_n \mid x_i \in \mathcal{V}\}$ to $y \in \mathbb{R}^{|\mathcal{V}|}$, which is a probability distribution over the next token continuation of $x$. Internally, $M$ has $L$ layers, where the output of the $\ell^{\text{th}}$ layer is computed as, $h^\ell = h^{\ell-1} + m^\ell + \sum_{j \leq J} a^{\ell j}$. Here, $m^\ell$ is the output of the MLP, and $a^{\ell j}$ is the contribution of $j^{\text{th}}$ attention head. For an individual head, its contribution to $h^\ell$ at token position $t$ is computed as:

$$a_t^{\ell j} = \sum_{i \leq t} h_i^{\ell-1} W_{OV}^{\ell j} \cdot \text{Attn}(q_t, K)_i \tag{1}$$

where $\quad q_t = h_t^{\ell-1} W_Q^{\ell j}, \quad K = h_{\leq t}^{\ell-1} W_K^{\ell j}, \quad$ and $\quad \text{Attn}(q_t, K) = \text{softmax}\left(\frac{q_t K^T}{\sqrt{d_{\text{head}}}}\right)$

Here, $\leq t$ denotes all tokens up to the current token $t$. Following Elhage et al. (2021), we combine the value projection $W_V^{\ell j}$ and out projection $W_O^{\ell j}$ in a single $W_{OV}^{\ell j}$. From here onward we will denote the $j^{\text{th}}$ attention head at layer $\ell$ as $[\ell, j]$.

**Filter Tasks.** In functional programming, the `filter` operation is used to select items from a collection that satisfy specific criteria. `filter` takes two arguments: the collection, and a *predicate* function that returns a boolean value indicating whether an item meets the criteria. Formally:

$$\texttt{filter}(\mathcal{C}, \psi) = \{c \in \mathcal{C} \mid \psi(c) \text{ is True}\} \tag{2}$$
$$\text{where} \quad \mathcal{C} = \{c_1, c_2, \ldots, c_n\} \text{ is a collection of items}$$
$$\text{and} \quad \psi : X \to \{\text{True}, \text{False}\} \text{ is the predicate function}$$

To study how language models implement filtering, we design a suite of filter-reduce tasks $\mathcal{T}$. For each task $\tau \in \mathcal{T}$, we construct a dataset $\mathcal{D}_\tau$ containing prompts $\{p_1, p_2, \ldots, p_m\}$. Each prompt $p_i = \mathbb{P}(\mathcal{C}, \psi)$ represents a natural language expression of a specific filter-reduce operation, where $\mathbb{P}$ denotes the verbalization function that converts the formal specification into natural language. Figure 1 shows a concrete example, and we include additional examples from each task in Appendix A.

## 2.2 FILTER HEADS

We observe that, for a range of filtering tasks, specific attention heads in the middle layers of Llama-70B consistently focus their attention on the items satisfying the given predicate, $\psi$. See Figure 1 (more in Appendix P) where we show the attention distribution for these filter heads from the last token position. From Equation (1), we know that this selective attention pattern emerges from the interaction between the query state at the last token ($q_{-1}$) and the key states from all preceding tokens $\left(K_{\leq t} = \{k_1, k_2, \ldots, k_t\}\right)$. We employ activation patching (Meng et al., 2022; Zhang & Nanda, 2024) to understand the distinct causal roles of these states.

To perform activation patching, we sample two prompts from $\mathcal{D}_\tau$: the source prompt, $p_{\text{src}} = \mathbb{P}(\mathcal{C}_{\text{src}}, \psi_{\text{src}})$ and the destination prompt, $p_{\text{dest}} = \mathbb{P}(\mathcal{C}_{\text{dest}}, \psi_{\text{dest}})$, such that the predicates are different ($\psi_{\text{src}} \neq \psi_{\text{dest}}$), and the collections are mutually exclusive ($\mathcal{C}_{\text{src}} \cap \mathcal{C}_{\text{dest}} = \emptyset$). We ensure that there is at least one item $c_{\text{targ}} \in \mathcal{C}_{\text{dest}}$, that satisfies $\psi_{\text{src}}$.

Figure 1 illustrates our activation patching setup with an example. For a filter head $[\ell, j]$ we analyze its attention pattern on three different forward passes.

**source run** $M(p_{\text{src}})$**:** We run the LM on the source prompt $p_{\text{src}}$ and cache the query state for $[\ell, j]$ at the last token position, $q_{-1}^{\ell j}$, hereafter denoted as $q_{\text{src}}$ for brevity.

**destination run** $M(p_{\text{dest}})$**:** The LM is run with $p_{\text{dest}}$.

**patched run** $M(p_{\text{dest}})[\leftarrow q_{\text{src}}]$**:** We run the LM with $p_{\text{dest}}$ again, but we replace the query state at the last token position for head $[\ell, j]$, $q_{-1}^{\ell j}$ with $q_{\text{src}}$ cached from the source run.

The attention patterns for the head $[\ell, j]$ from the three forward passes for an example prompt pair are depicted in Figure 1(b), (e), and (f) respectively. In the source and destination runs, the head attends to the items that satisfy the respective predicates. But in the patched run, the filter head $[\ell, j]$ shifts its attention to the item in $\mathcal{C}_{\text{dest}}$ that satisfies $\psi_{\text{src}}$. Patching $q_{\text{src}}$ is enough to trigger the execution of $\psi_{\text{src}}$ for this head in a different context, validating that $q_{\text{src}}$ encodes a compact representation of $\psi_{\text{src}}$.

Notably, we cache the query states before the positional embedding (Su et al., 2024) is applied, while $\text{Attn}(q_t, K)$ in Equation (1) is calculated after the position encoding is added. This indicates that filter heads are a category of semantic heads (Barbero et al., 2025) with minimal sensitivity to the positional information.

## 2.3 LOCATING FILTER HEADS

Now we introduce the methodology to systematically locate these filter heads within a LM.

**Activation Patching with Learned Masks.** While analyzing attention patterns can provide valuable insights, attention patterns can sometimes be deceptive (Jain & Wallace, 2019) as they may not give insights into the underlying *causal* mechanisms of the LM (Grimsley et al., 2020). To address this issue, we perform causal mediation analysis with the activation patching setup discussed in

Section 2.2 to identify the heads carrying the predicate representation. We want to find a set of heads that *causes* the score (logit or probability) of the target item $c_{\text{targ}}$ to increase in the patched run.

We begin by patching the attention heads *individually* and selecting the heads that maximize the logit difference of $c_{\text{targ}}$ in the patched run vs the destination run, $\text{logit}[\leftarrow q_{\text{src}}](c_{\text{targ}}) - \text{logit}(c_{\text{targ}})$. We use logits instead of probabilities as logits have a more direct linear relationship with the influence caused by the intervention (Zhang & Nanda, 2024).

However, we find that patching a single filter head is often not a strong enough intervention to exert influence over the final LM behavior because other filter heads, in addition to backup mechanisms (Wang et al., 2023; McGrath et al., 2023), may work against the intervention and rectify its effects. To address this issue, we learn a sparse binary mask over all the attention heads, similar to De Cao et al. (2020) and Davies et al. (2023). We cache the query states for the source run $M(p_{\text{src}})$ and destination run $M(p_{\text{dest}})$, and then perform the following interchange intervention over the query states of all the attention heads in the patched run:

$$q_{-1}^{\ell j} \leftarrow \text{mask}^{\ell j} * q_{\text{src}}^{\ell j} + (1 - \text{mask}^{\ell j}) * q_{\text{dest}}^{\ell j} \tag{3}$$

Here, the vector $q_{-1}^{\ell j}$ denotes the query state of the $j$th head at layer $\ell$ at the last token position; and $q_{\text{src}}^{\ell j}$ and $q_{\text{dest}}^{\ell j}$ are the query states of that same head at the last token from $M(p_{\text{src}})$ and $M(p_{\text{dest}})$, respectively. The terms $\text{mask}^{\ell j}$ are each binary values, jointly learned with an objective to maximize the logit of $c_{\text{targ}}$, while suppressing other options in $C_{\text{dest}}$, in the patched run. We use a sparsity regularizer to ensure that the mask is sparse (i.e., only a few heads are selected). In Figure 1(g) we mark the filter heads identified for one of our filtering tasks, *SelectOne — Obj*, with their individual average indirect effect (AIE) of promoting the logit of $c_{\text{targ}}$.

**Causality.** If the identified filter heads fully capture a compact representation of the predicate $\psi$ in their query states which is used by the LM to perform the filtering operation, then transferring $q_{\text{src}}$ from $M(p_{\text{src}})$ to $M(p_{\text{dest}})$ should be *causally* influential: it should cause the LM to select $c_{\text{targ}}$, the item in $C_{\text{dest}}$ that satisfies $\psi_{\text{src}}$. We introduce a *causality* score to quantify the collective causal influence of the selected filter heads.

$$c^* = \underset{c \in C_{\text{dest}}}{\text{argmax}} \; \left( M(p_{\text{dest}}) \left[ q_{-1}^{\ell j} \leftarrow q_{\text{src}}^{\ell j} \mid \forall [\ell, j] \in \mathcal{H} \right] \right)_t$$

$$\text{Causality}(\mathcal{H}, p_{\text{src}}, p_{\text{dest}}) = \mathbb{1}\left[ c^* \overset{?}{=} c_{\text{targ}} \right] \tag{4}$$

$$\text{where} \quad \mathcal{H} \text{ is the set of all selected filter heads}$$

We run the LM on $p_{\text{dest}}$ and patch the query state of only the selected heads at the last token position with their corresponding query states cached from $M(p_{\text{src}})$. We then check if the LM predicts $c_{\text{targ}}$ as the most probable item in the LM's output distribution among all items in $C_{\text{dest}}$.

Notably, while finding the heads we do not care if the heads exhibit the attention behavior illustrated in Figure 1. But, we notice that the aggregated attention pattern of the identified heads consistently aligns with the selective attention pattern for filtering (see Appendix P for some examples).

## 3 EXPERIMENTS

We now empirically test the role of filter heads in different settings to validate our claims.

**Models.** We study autoregressive transformer LMs in our experiments. Unless stated otherwise, all the reported results are for Llama-70B (Touvron et al., 2023). We include additional results for Gemma-27B (Team et al., 2024) in Appendix L.

**Datasets.** To support our evaluation, we curate a dataset consisting of six different tasks that all require the LM to perform filtering, followed by a reduce step to provide a specific answer. Each task-specific dataset $\mathcal{D}_\tau$ contains a collection of items categorized in different categories (e.g. *fruits*, *vehicles*, ... in *Obj type*), as well as different prompt templates for questions specifying the predicate and the reduction task (e.g. *How many [category]s are in this list?*). When we curate a prompt for the task, we sample the collection from the items in $\mathcal{D}_\tau$ and fill in the template with the target predicate. The tasks are listed in Figure 3, and see Appendix A for example prompts from each task.

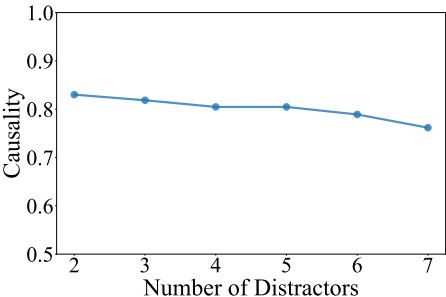

Figure 2: Filter heads retain a causality close to 0.8 even with 7 distractors in the collections of the destination prompt.

Table 1: Causality of filter heads on *SelectOne* tasks. Heads identified using object-type filtering (e.g., *find the fruit*) generalize to semantically distinct predicates like profession identification (*find the actor*).

| Semantic Type | Causality | $\Delta$logit |
|---|---|---|
| Object Category | 0.863 | +9.03 |
| Person Profession | 0.836 | +7.33 |
| Person Nationality | 0.504 | +5.04 |
| Landmark in Country | 0.576 | +7.02 |
| Word rhymes with | 0.041 | +0.65 |

**Implementation Details.** For each task we locate the filter heads using the method detailed in Section 2.3 on 1024 examples. During localization we perform the interchange operation (Equation (3)) only at the last token, but for evaluation we consider last 2 tokens ({"\nAnswer", ":" }) to reduce information leakage. We also calculate $q_{src}$ as a mean of $n$ source prompts achieved from a single $p_{src}$ by changing the index of $c_{src}$ in $\mathcal{C}_{src}$[1]. While sampling the counterfactual prompts, we ensure that the answer for the source prompt, destination prompt, and the target answer for the patched prompt are all different from each other. All the reported scores are evaluated on a draw of 512 examples where the LM was able to correctly predict the answer. In some cases we include $\Delta$logit, the logit difference of $c_{targ}$ in the patched run versus the destination run, as a softer metric to causality from Equation (4).

## 3.1 PORTABILITY/GENERALIZABILITY WITHIN TASK

Following the approach detailed in Section 2.3, we identify the filter heads on the *SelectOne* task for object categorization. While localizing these heads we use English prompts that follow a specific format: the items are presented in a single line and the question specifying the predicate is presented *after* the items. We test whether the filter heads identified with this format generalize to various linguistic perturbations and *SelectOne* tasks that require reasoning with information of different semantic type. We evaluate generalization using the causality score (Equation (4)).

**Information Types.** Table 1 shows that filter heads identified on object categorization maintain high causality even in entirely different semantic domains — notably, identifying people by profession shows comparable causality despite the semantic shift. The filter heads also retain non-trivial causality for person-nationality and landmark-country associations, with causality improving by approximately 10 points when we include prefixes which prime the LM to recall relevant information in the item representations (see Appendix G).

However, the predicates captured by these filter heads show poor causality in situations that require reasoning with non-semantic information, such as identifying rhyming words. This indicates that the filter heads play a causal role specifically in situations that require filtering based on semantic information rather than non-semantic properties like phonological similarity or letter counting.

**Size of the collection, $\mathcal{C}$** In Figure 2 we plot the causality of filter heads by varying the number of distractor items in the list. The figure shows that the heads are not very sensitive to the size of the collection, retaining high causality even with 7 distractors.

**Linguistic Variations.** Recent works have shown that LLMs capture language-independent abstractions in their internal representations (Dumas et al., 2024; Feucht et al., 2025). Building on these findings we test whether predicate representations in filter heads remain causal under linguistic perturbations by extracting $q_{src}$ from one prompt format and applying it to destination prompts with different presentation styles, or even languages. Table 2(a) and (b) demonstrate remarkable robustness: the same filter heads maintain high causality across different item presentation formats, and even cross-lingual transfer. This invariance to surface-level variation confirms that filter heads encode abstract semantic predicates rather than pattern-matching on specific linguistic forms. However, we also observe that when the question is presented *before* the items, the filter heads show poor causality (see Table 2(c)). We find that this is because in the question-before case the LM relies more on a complementary implementation of filtering, which we discuss in Section 5 and in Appendix B. All the other results presented in this section are calculated on prompts following the question-after format.

---

[1]This slightly increases the causality by removing the order information. See Appendix F.

Table 2: Portability of predicate representations across linguistic variations. The predicate vector $q_{src}$ is extracted from a source prompt and patched to destination prompts in **(a)** different languages, **(b)** different presentation formats for the items, and **(c)** placing the question before or after presenting the collection.

| **From** | English | Spanish | French | Hindi | Thai |
|---|---|---|---|---|---|
| English | **0.863** | 0.893 | 0.779 | 0.928 | 0.951 |
| Spanish | 0.857 | **0.877** | 0.775 | 0.875 | 0.891 |
| French | 0.938 | 0.932 | **0.793** | 0.931 | 0.9473 |
| Hindi | 0.920 | 0.920 | 0.885 | **0.918** | 0.957 |
| Thai | 0.897 | 0.928 | 0.887 | 0.940 | **0.943** |

**(a)** Cross-lingual transfer

| **From** | single line | bulleted |
|---|---|---|
| single line | **0.863** | 0.842 |
| bulletted | 0.840 | **0.848** |

**(b)** Across option presentation style

| **From** | after | before |
|---|---|---|
| after | **0.863** | 0.580 |
| before | 0.398 | **0.020** |

**(c)** Placement of the question

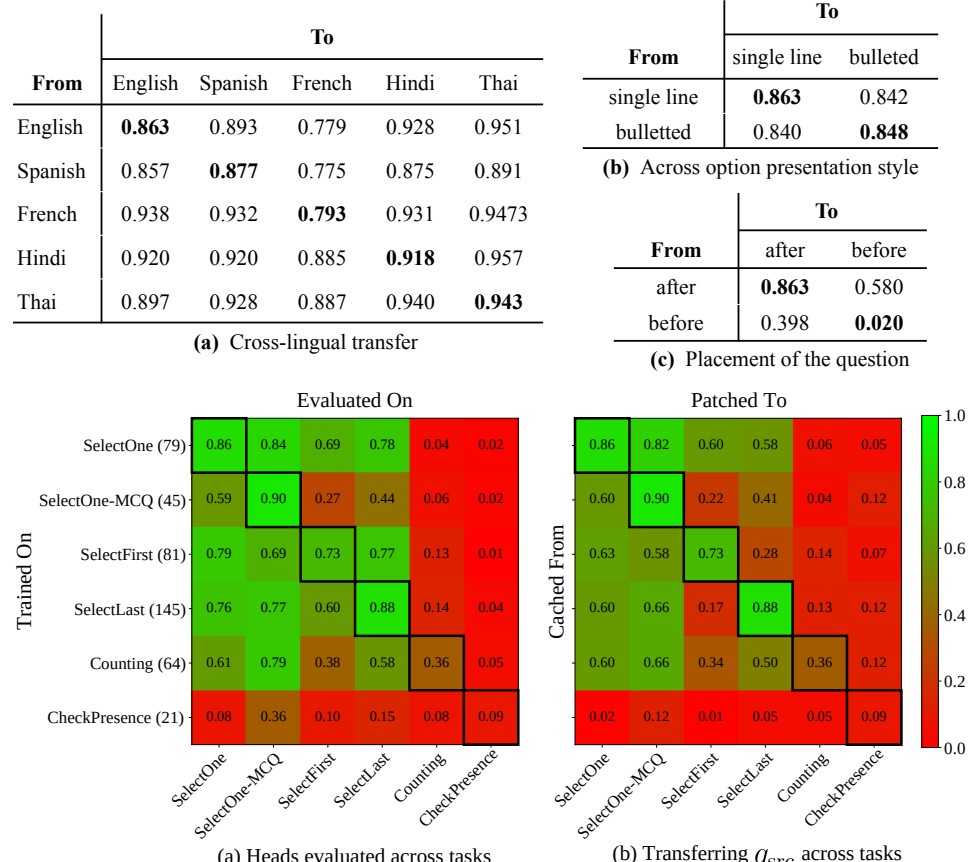

(a) Heads evaluated across tasks    (b) Transferring $q_{src}$ across tasks

Figure 3: Generalization across different tasks. **(a)** shows whether the heads identified with one task (rows) maintain causal influence in another task (columns). **(b)** shows how portable the predicate representation is across tasks. The predicate rep $q_{src}$ is cached from one source task example (e.g., *find the fruit* in SelectOne task) and is patched to an example from another destination task (e.g., *count the vehicles* in Counting task). The heatmap shows causality scores, i.e. whether the LM correctly performs the destination task with the transferred predicate (e.g., *count the fruits*). For both **(a)** and **(b)** the values in the diagonal grid show within task scores.

## 3.2 PORTABILITY/GENERALIZABILITY ACROSS FILTER-REDUCE OPERATIONS

To understand the scope of filter head usage, we examine their participation across six filter-reduce tasks of different complexity. Each of these tasks requires the LM to perform a separate reduce step to produce an answer in a specific format. We measure whether the heads identified from one task maintain their causality when tested on another task.

Figure 3(a) reveals two distinct patterns. First, transferring the heads across the four tasks — *SelectOne*, *SelectOne(MCQ)*, *SelectFirst*, and *SelectLast* — shows high causality scores ($\geq 70\%$[2]) among them, which indicate a high overlap of the same filter heads. In contrast, *Counting* shows an interesting asymmetric pattern: while *Select\** heads fail on the *Counting* task, *Counting* heads show partial generalization to the *Select\** tasks — suggesting that *Counting* does share some common sub-circuit with *Select\** tasks, while having a more complex mechanism, likely involving additional circuits for specialized aggregation, that we have not yet identified. *CheckPresence* heads show poor causality even within the task, indicating that the LM possibly performs this task in an alternate way that can bypass the filtering sub-circuit.

We also test the portability of the predicate information encoded in $q_{src}$ by transferring it across tasks, Figure 3(b). *SelectFirst* and *SelectLast* tasks show notably poor cross-task transfer of the predicate, even though the filter heads retain high within-task causality. This suggests that, in $q_{src}$ the predicate

---

[2]Except the heads from *SelectOne-MCQ*, possibly because selecting MCQ-options is computationally simpler than to output the filtered item.

Table 3: LM performance on filtering tasks drops significantly when filter heads are ablated. These heads constitute $< 2\%$ of the heads in the LM. Evaluated on 512 samples that the LM predicts correctly without any ablation (baseline 100%).

| Task(#Heads) | LM Acc (Heads Abl) | |
| --- | --- | --- |
| | Filter | Random |
| SelectOne (79) | 22.5% | 99.6% |
| SelectOne(MCQ) (45) | 0.4% | 100% |
| SelectFirst (81) | 13.1% | 97.3% |
| SelectLast (145) | 9.22% | 99.4% |
| Count (64) | 89.80% | 99.19% |
| CheckExistence (21) | 98.61% | 99.2% |

Table 4: Filter heads play a distinct causal role during filtering tasks. The table shows the causality of filter heads with other types of heads documented in the literature. None of the other head types match the causality of filter heads in the *SelectOne* task. To keep our comparisons fair, we keep the number of heads equal (79) for every head type.

| Head Type | Causality | $\Delta$logit |
| --- | --- | --- |
| Filter | 0.863 | $+9.03$ |
| Function Vector | 0.002 | $-2.13$ |
| Concept Induction | 0.080 | $+5.23$ |
| Token Induction | 0.00 | $-3.23$ |
| Random | 0.00 | $-0.96$ |

information is possibly entangled with task-specific information. Otherwise, predicate transfer scores (Figure 3b) mirror the head transfer scores (Figure 3a) with slightly lower values.

Our findings suggest that filter heads form a foundational layer for a range of reduce operations, with simpler selection tasks relying primarily on this mechanism while more complex aggregation tasks build additional computation on top of it. This insight aligns with Merullo et al. (2024a) that transformer LMs use common sub-circuits (filter heads) across different (filter-reduce) tasks.

### 3.3 NECESSITY OF FILTER HEADS

We seek to understand to what extent the LM relies on filter heads during these filter-reduce tasks. To assess their importance, we perform ablation studies.

We ablate an attention head $[\ell, j]$ by modifying its attention pattern during the forward pass so that the last token can only bring information from the <BOS> token[3]. Previous works (Geva et al., 2023; Sharma et al., 2024) have investigated if critical information flows through a certain attention edge with similar attention knock-out experiments. The results in Table 3 reveal a dramatic performance drop for the *Select\** tasks when filter heads are ablated, despite these heads comprising less than 2% of the model's total attention heads — confirming their critical importance for the *Select\** tasks. In contrast the performance for *Counting* and *CheckExistence* do not drop significantly due to this ablation, again indicating that these tasks do not fully rely on the filter heads.

To determine whether filter heads represent a novel discovery or merely overlap with existing attention head categories previously documented in the literature, we compared their functionality against such head categories with specific functional roles. Specifically, we measure the causality of Function Vector heads (Todd et al., 2024), Concept Induction heads (Feucht et al., 2025), and Token Induction heads (Olsson et al., 2022). As shown in Table 4, none of these previously identified head types exhibit the distinctive functional role of filter heads. We do not treat these heads as "baselines" in the traditional sense, as they are not intended for filtering tasks. We evaluate them to confirm that filter heads are a unique and previously unrecognized component in transformer LMs.

We also verify that filter heads encode abstract predicates rather than specific answer tokens: even when source prompts contain the predicate but no valid answer item in the collection, causality remains high at 0.80 (see Appendix E). Moreover, these predicate representations support composition through vector arithmetic: adding the query vectors of two predicates (e.g., *is_fruit* and *is_vehicle*) produces a vector that executes their disjunction, achieving a causality of 0.65 (see Appendix D). These results further confirm that filter heads encode predicates as compact geometric directions in query space.

## 4 KEY STATES CARRY ITEM SEMANTICS FOR FILTERING

To understand *how* the predicate-encoding query states in the filter heads implement filtering via interacting with the key states from previous tokens, we design another activation patching experiment.

**Approach.** To isolate the contribution of key states, we designed a two-part intervention that combines query patching with key swapping. We select two items $(c_{\text{targ}}, c_{\text{other}})$ from $\mathcal{C}_{\text{dest}}$ such that $c_{\text{targ}}$ satisfies $\psi_{\text{src}}$, but not $\psi_{\text{dest}}$; and $c_{\text{other}}$ doesn't satisfy either of the predicates.

---

[3]for the LM tokenizers that do not add a <BOS> token by default, we prepend <BOS> manually.

The intervention proceeds as follows. For a filter head $[\ell, j]$, we patch the $q_{\text{src}}$ from the source prompt (as before). Additionally, we swap the key states between $c_{\text{targ}}$ and $c_{\text{other}}$ within the same forward pass. Figure 4 illustrates this key-swapping. After this additional key-swapping intervention, the filter head $[\ell, j]$ redirects its focus from $c_{\text{targ}}$ ( Figure 1(f)) to $c_{\text{other}}$ (Figure 4-right).

**Result.** We consider this 2-step intervention to be causally effective if the LM assigns the highest probability to $c_{\text{other}}$ among the items. On the *SelectOne* object categorization task, we achieve a causality score of 0.783 (432/552 examples) with $\Delta\text{logit} = 8.2591 \pm 3.352$, confirming that key states indeed encode the semantic properties that predicates evaluate. For experimental simplicity, we restrict this analysis to only single-token items.

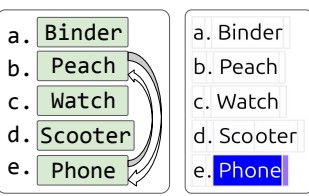

Figure 4: Swapping the key states of the items in addition causes the filter head to redirect its focus to an unrelated item.

These results confirm our mechanistic hypothesis: filter heads implement filtering through a localized key-query interaction where queries selectively encode *what to look for* (the predicate) while discarding the item content and other unrelated prompt context, and key states similarly bring *what is in the list* by extracting the semantic properties of the items from the corresponding latents ($h^{\ell-1}$), discarding the surrounding context. Notably, this behavior manifests in the attention pattern. Although previous works cautioned that attention patterns can be misleading (Jain & Wallace, 2019), in our predicate-matching setting we see a clear correspondence between how filter heads utilize abstract predicate encodings to attend to what was asked for and how they causally influence the LM behavior — doing what Vaswani et al. (2017) and the original designers of attention might have envisioned.

## 5 WHAT HAPPENS IF THE QUESTION COMES *before* THE OPTIONS?

In Table 2(c) we see that if we simply reverse the order of the question and the collection to ask the question *before* presenting the items, the causality scores drop to near zero. Our investigations reveal that this seemingly innocent ordering change fundamentally alters the computational strategy available to the LM.

**Hypothesis.** When the question comes first, the transformer can perform *eager evaluation*: as each item is processed, the model can immediately evaluate whether it satisfies the predicate and store this information as an *is_match* flag directly in the item's latents. And, at the final token position, rather than performing the predicate matching operation via filter heads, the LM simply retrieves items based on these pre-computed flags.

If this hypothesized flagging mechanism is true, then manipulating this flag should result in predictable outcomes in the LM's behavior. We find evidence for this alternative mechanism through a series of carefully designed activation patching experiments. We illustrate the most decisive experiment setup in Figure 5, while we leave the detailed analysis to Appendix B.

**Experiment Setup.** Figure 5 illustrates our two-part causal intervention setup to test for the presence of the hypothesized flag information in the item latents. Similar to the setup described in Section 2.2, we consider two prompts, a source prompt $p_{\text{src}} = \mathbb{P}(\mathcal{C}, \psi_{\text{src}})$ and a destination prompt $p_{\text{dest}} = \mathbb{P}(\mathcal{C}, \psi_{\text{dest}})$, both following either the question-before or question-after format. As in Section 2.2, the predicates are different ($\psi_{\text{src}} \neq \psi_{\text{dest}}$), but they operate on the same collection of items in both the source and the destination prompt ($\mathcal{C}_{\text{src}} = \mathcal{C}_{\text{dest}} = \mathcal{C}$). Let $c_{\text{src}}$ and $c_{\text{dest}}$ be the items in $\mathcal{C}$ satisfying the predicates $\psi_{\text{src}}$ and $\psi_{\text{dest}}$ respectively. Our goal is to engineer a situation where a third item $c_{\text{flag}} \notin \{c_{\text{src}}, c_{\text{dest}}\}$ is the sole carrier of the *is_match* flag, and then check whether the LM is misled into selecting $c_{\text{flag}}$.

We achieve this with two simultaneous interventions:

(a) **Transferring the predicate/decision context.** In the destination run with $p_{\text{dest}}$, we patch the *residual* states of a *single* layer $\ell$, at the final two token positions ("\nAns" and ":") from the source run. When applied in later layers this intervention simply copies the final decision from the source run. But patching early layers transfers the predicate. See Figures 7 and 11 in Appendix B for details.

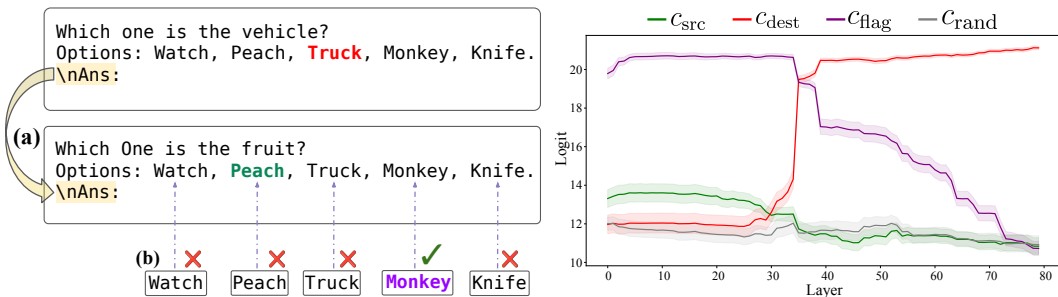

Figure 5: Testing for eagerly computed answer flags in question-before prompts. **Left:** Two-part intervention setup. **(a)** We patch the residual states at the final two token positions (`"\nAns"` and `":"`) from a source prompt $p_{\text{src}}$ to a destination prompt $p_{\text{dest}}$ at a single layer. **(b)** Additionally, for an item $c \in \mathcal{C}_{\text{dest}}$, we replace its hidden representations across *all* layers with those from a prompt $p_{\text{diff}}$ containing a different predicate $\psi_{\text{diff}} \notin \{\psi_{\text{src}}, \psi_{\text{dest}}\}$. We use a different $p_{\text{diff}}$ (with different $\psi_{\text{diff}}$) per item in $\mathcal{C}_{\text{dest}}$. Crucially, we ensure that exactly one item $c_{\text{flag}}$ (distinct from both the source and destination answers) carries the *is_match* flag from its corresponding $p_{\text{diff}}$. **Right:** Logits of tracked items on the *SelectOne-Obj* task as $\ell$ varies for **(a)**, with **(b)** applied throughout. After this 2-step intervention the LM consistently selects $c_{\text{flag}}$ in early layers, confirming the LM's reliance on pre-computed answer flags stored in the item latents.

(b) **Planting the is_match flag on another item.** We construct another prompt $p_{\text{flag}} = \mathbb{P}(\mathcal{C}, \psi_{\text{flag}})$, with a predicate $\psi_{\text{flag}} \notin \{\psi_{\text{src}}, \psi_{\text{dest}}\}$ that is satisfied by the third item $c_{\text{flag}} \in \mathcal{C}\backslash\{c_{\text{src}}, c_{\text{dest}}\}$. In $M(p_{\text{flag}})$, only $c_{\text{flag}}$ carries the *is_match* flag. To plant the flag on $c_{\text{flag}}$ in the patched run, we replace the item's latents with corresponding latents cached from $M(p_{\text{flag}})$ across *all* layers.

Similarly to ensure that an item $c' \in \mathcal{C}\backslash\{c_{\text{flag}}\}$ does not carry the flag, we construct another prompt $p' = \mathbb{P}(\mathcal{C}, \psi')$ such that $\neg\psi'(c')$. We then swap the latents of $c'$ for *all* layers in the patched run with the corresponding latents from $M(p')$. Doing these replacements for $c' \in \mathcal{C}\backslash\{c_{\text{flag}}\}$ ensures that only $c_{\text{flag}}$ carries the flag in the patched run. See Figure 10 in Appendix B for a concrete example of $p_{\text{flag}}$ and $p'$.

**Results.** Figure 5 (right panel) shows the logit of different items as we vary the layer $\ell$ at which intervention **(a)** is applied, while intervention **(b)** is applied across all the layers throughout. In the question-before setting, the LM consistently selects $c_{\text{flag}}$ in early-to-middle layers, which is precisely the behavior predicted by the flag-based hypothesis. The LM ignores both the source predicate ($\psi_{\text{src}}$) and the destination predicate ($\psi_{\text{dest}}$), and follows the planted flag instead. In contrast, replicating this experiment in the question-after setting shows that the LM is not strongly influenced by the flag, which is expected as the LM didn't get an opportunity to precalculate the flag in the item latents when the predicate is presented after the items. See Appendix B for details on this.

**Filter heads remain partially active.** Despite the dominance of the flag-based mechanism in the question-before setting, filter heads are not entirely inactive. Table 2(c) shows that caching $q_{\text{src}}$ from question-before prompts and patching to question-after prompts yields non-trivial causality (0.398), indicating that filter heads still partially encode predicate information in question-first prompts. The two mechanisms thus operate in parallel rather than in mutual exclusion, with the flag-based pathway typically dominating when the question precedes the collection.

## 6 APPLICATION: A LIGHT-WEIGHT PROBE WITHOUT TRAINING

The predicate information encoded by the filter heads can be leveraged for a practical use-case: zero-shot concept detection through training-free probes.

Since filter heads encode predicates as query states that interact with key-projected item representations to perform filtering, we can repurpose this mechanism for classification. To detect whether a representation $h$ belongs to a particular concept class (e.g., animal, vehicle), we create filter prompts for each class: $p_{\text{cls}} = \mathbb{P}(\mathcal{C}, \psi_{\text{is\_cls}})$ and collect the query states $q_{\text{cls}}$ from a filter head $[\ell, j]$.

Then we classify $h$ by finding the class whose $q_{\text{cls}}$ has the maximum affinity.

$$\hat{y} = \arg\max_{\text{cls}} \left( q_{\text{cls}} \cdot W_K^{\ell j} h \right) \tag{5}$$

Where $W_K^{\ell j}$ is the key projection from the head $[\ell, j]$. Figure 6 demonstrates that this approach achieves strong classification performance without any training, validating that filter heads learn generalizable concept representations that can be extracted and applied as probes.

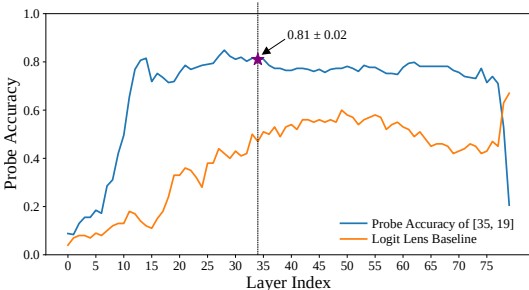

Filter heads encode predicates in compact geometric directions that produce distinctive attention patterns via their interaction with the key states. We can use this insight to utilize filter heads to identify false information or detect sentiment in free-form text. See Appendix P.4 for some examples.

Figure 6: Training-free probe using filter head [35, 19]. Accuracy across layers on 238 objects spanning 16 classes from the SelectOne-Object dataset. Final token is used for multi-token items. Compared against using the embedding vectors of LM decoder as class probes.

## 7 RELATED WORKS

**Attention Head Studies.** Previous works have identified specialized attention heads that serve distinct computational roles. Olsson et al. (2022) discovered induction heads that implement pattern matching and copying, while Feucht et al. (2025) have identified heads that copy concepts instead of individual tokens. Todd et al. (2024) have found function vector heads that encode task representations that are transportable across contexts. Filter heads are an addition to this class of attention heads that show distinct functional specialization.

**LM Selection Mechanisms.** A few empirical studies have explored the selection mechanism in LMs, primarily in MCQA settings. Tulchinskii et al. (2024) identifies "select-and-copy" heads based on their attention pattern that focus on "\n" after a correct item in a question-first MCQ format. Lieberum et al. (2023) also identify attention heads that attend to the correct MCQ label/letter and show that these "correct label" heads encode the ordering ID of the presented options. Wiegreffe et al. (2025) showed that attention modules in the middle layers promote the answer symbols in a MCQA task. Unlike these works focused on MCQA settings, in this paper we investigate list-processing in general and find a set of filter heads that implement predicate evaluation that generalize across formats, languages, and even different reduction operations.

**Symbolic Reasoning in Neural Networks.** Recently researchers have been increasingly interested in the question of whether transformer LMs can develop structed symbolic-style algorithmic behavior. Yang et al. (2025) discuss how LMs can implement an abstract symbolic-like reasoning through three computational stages: with early layers converting tokens to abstract variables, middle layers performing sequence operations over these variables, and then later layers accessing specific values of these variables. Meng et al. (2022) and Geva et al. (2023) also notice similar stages while the LM recalls a factual association. Several works have documented mechanisms/representations specialized for mathematical reasoning (Nanda et al., 2023; Hanna et al., 2023; Kantamneni & Tegmark, 2025) and variable binding (Feng & Steinhardt, 2024; Prakash et al., 2025).

Our paper continues this tradition of validating Smolensky (1991)'s assertion that distributed representations in connectionist systems can have "sub-symbolic" structures, with symbolic structures emerging over the interaction between many units. In this work we study a specific symbolic abstraction — filtering in list processing — which is a fundamental abstraction for both symbolic computation and human reasoning (Treisman, 1964; Johnson-Laird, 1983).

## 8 DISCUSSION

In this work, we have identified and characterized filter heads — specialized attention heads that implement filtering operations in autoregressive transformer LMs. These heads encode the filtering criteria (predicates) as compact representations in their query states of specific tokens. This encoding can be extracted and then transported to another context to trigger the same operation. We also identify that, based on information availability, the LM can use an *eager* implementation of filtering by storing flags directly on the item latents. These dual and complimentary filtering implementations mirror the lazy vs eager evaluation from functional programming. This convergence between emergent neural mechanisms and human-designed programming primitives suggests that certain computational patterns arise naturally from task demands rather than architectural constraints.

ETHICS

This research investigates the internal computational mechanisms of LMs through mechanistic interpretability techniques, contributing to the scientific understanding of transformer architectures. Our identification of filter heads advances LMs transparency by revealing how models implement functional programming primitives, though we acknowledge that interpretability findings do not directly translate to safety improvements without additional work. The causal mediation techniques we develop could potentially be applied to study more sensitive model capabilities, requiring responsible application and consideration of dual-use implications in future research on mechanisms related to deception or manipulation. Our experiments require significant computational resources that may limit reproducibility to well-resourced institutions, though we commit to releasing code and datasets to facilitate broader access. While our findings about filter heads appear robust across different tasks and languages, we caution against overgeneralizing to other domains without validation, as mechanistic interpretability remains early-stage and our understanding of component interactions is incomplete.

REPRODUCIBILITY

The code and dataset produced in this work are available at `github.com/arnab-api/filter`. We ran all experiments on workstations with either 80GB NVIDIA A100 GPUs or 48GB A6000 GPUs, using the HuggingFace Transformers library (Wolf et al., 2019) and PyTorch (Paszke et al., 2019). We used NNsight (Fiotto-Kaufman et al., 2025) for our intervention experiments.

ACKNOWLEDGEMENTS

This research has been supported by a grant from Open Philanthropy (DB, AS, NS), and the Israel Council for Higher Education (NS). AS received compute credits from the U.S. NSF Advanced Cyberinfrastructure Coordination Ecosystem: Services & Support (NSF ACCESS), which supported some of our experiments.

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

# A    EXAMPLE PROMPTS FROM OUR DATASET

## A.1    DIFFERENT TASKS

### Select One – Type of Object

"Options: Bus, Peach, Scooter, Phone, Pen
Find the fruit in the options presented
above.
Answer:"

Expected LM Output: " Peach"

### Select One – Type of profession

"Options: Neymar, Hillary Clinton, Clint
Eastwood
Who among these people mentioned above is
an actor by profession?
Answer:"

Expected LM Output: " Clint"

### Select One – Type of nationality

"Options: Ronaldinho, Brad Pitt, Jet Li,
Ken Watanabe
Who among these people mentioned above is
from China?
Answer:"

Expected LM Output: " Jet"

### Select One – Location of landmark

"Options: Cabo San Lucas Arch, Plaza de
Armas Cusco, Mont Saint-Michel
Which of these landmarks is in Peru?
Answer:"

Expected LM Output: " Plaza"

### Select One — Rhyme

"Options: blue, debt, bright, sting, sake
Which of these words rhymes with glue?
Answer:"

Expected LM Output: " blue"

### Select One (MCQ)

"a. Banana
b. Paperclip
c. Oven
d. Dress
e. Church
f. Bench
Which among these objects mentioned above
is a clothing?
Answer:"

Expected LM Output: " d"

### Select First

"Options: Church, Scarf, Pendant, Slow
cooker, Temple
What is the first building from the list
above?
Answer:"

Expected LM Output: " Church"

### Select Last

"Options: Horse, Anklet, Golf ball, Cow,
Necklace
What is the last animal in this list
above?
Answer:"

Expected LM Output: " Cow"

### Counting

"Options: Trombone, Flute, Guitar, Train,
Car
How many vehicles are in this list?
Answer:"

Expected LM Output: " Two"

### Check Existence

"Options: Refrigerator, Museum, Notebook,
Toaster, Juicer
Do you see a kitchen appliance in the list
above?
Answer:"

Expected LM Output: " Yes"

## A.2 LINGUISTIC PERTURBATIONS

### A.2.1 ITEM PRESENTATION

**Single Line**

"Options: House, Blender, Willow, Truck,
Piano, Wrestling mat.
Which among these objects mentioned above
is a kitchen appliance?
Answer:

Expected LM Output: " Blender"

**Bulleted**

"* Temple
* Air fryer
* Basketball
* Willow
* Van
* Harmonica
Which among these objects mentioned above
is a vehicle?
Answer:"

Expected LM Output: " Van"

### A.2.2 QUESTION PLACEMENT

**Question After**

"Options: Elephant, Maple, Toilet, Camera,
Juicer, Mall.
Which among these objects mentioned above
is a bathroom item?
Answer:"

Expected LM Output: " Toilet"

**Question Before**

"Which object from the following list is
a music instrument?
Options: Printer, Highlighter, Ukulele,
Chair, Mirror, Locket.
Answer:"

Expected LM Output: " Uk"

### A.2.3 FROM A DIFFERENT LANGUAGE

**Spanish**

"Opciones: Lirio, Colchoneta de lucha,
Escritorio, Portátil, Refrigerador,
Sandía.
¿Cuáles de estos objetos mencionados
anteriormente son un(a) electrónica?
Respuesta:"

Expected LM Output: " Port"

**French**

"Options : Aigle, Pastèque, Accordéon,
Baignoire, Ciseaux, Bibliothèque.
Lequel de ces objets mentionnés ci-dessus
est un(e) fourniture de bureau ?
Réponse :"

Expected LM Output: " d"

**Hindi**

"विकल्प: शेर, साबुन, टेनिस बॉल, अंगूर, मिक्सर, बस.
उपरोक्त वस्तुओं में से कौन-सी एक जानवर है?
उत्तर:"

Expected LM Output: " श"

**Thai**

"ตัวเลือก: หัวผักกาด, สิงโต, แดฟโฟดิล, กาต้มน้ำ, ชุดเดรส, เชลโ�่.
วัตถุใดในรายการข้างต้นที่เป็น เสื้อผ้า?
คำตอบ:"

Expected LM Output: " ช"

## A.3 LM BASELINE PERFORMANCE ON TASKS

Table 5: LLM baseline performance on filtering tasks. All the models were evaluated on the same draw of 1024 examples for each of the tasks. The LMs are sorted in descending order by size. The question was presented after the items.

| Model (Huggingface ID) | SelectOne | SelectOne-MCQ | SelectFirst | SelectLast | Counting | CheckPresence |
|---|---|---|---|---|---|---|
| Llama-70B (meta-llama/Llama-3.3-70B-Instruct) | 99.32% | 98.63% | 78.42% | 87.01% | 67.38% | 91.02% |
| Gemma-27B (google/gemma-2-27b-it) | 99.02% | 97.75% | 75.29% | 88.96% | 64.16% | 96.29% |
| Gemma-9B (google/gemma-2-9b-it) | 98.93% | 97.85% | 68.55% | 78.03% | 63.67% | 95.90% |
| Llama-8B (meta-llama/Llama-3.1-8B-Instruct) | 98.44% | 97.95% | 50.88% | 68.65% | 63.28% | 94.63% |

Table 6: LM baseline accuracy when the question is presented before vs after the items. Reported for *SelectOne* task for object categorization. All LMs were evaluated on the same draw of 1024 examples varying the question placement.

| Models | Ques Placement | |
|---|---|---|
| | Before | After |
| Llama-70B | 99.32% | 99.32% |
| Gemma-27B | 99.41% | 99.02% |
| Gemma-9B | 98.93% | 99.61% |
| Llama-8B | 99.51% | 98.44% |

Table 7: LM baseline accuracy for the SelectOne tasks that require dealing with different semantic information. All the LMs evaluated on the same draw of 1024 examples per semantic type. The question was presented after the options here.

| Models | Semantic Types | | | |
|---|---|---|---|---|
| | Object | Profession | Nationality | Landmark-country |
| Llama-70B | 99.32% | 98.44% | 99.22% | 100.00% |
| Gemma-27B | 99.02% | 98.93% | 91.60% | 98.14% |
| Gemma-9B | 98.93% | 98.73% | 96.78% | 98.14% |
| Llama-8B | 98.44% | 98.14% | 94.04% | 98.24% |

## B    DUAL IMPLEMENTATION OF FILTERING IN LMS: QUESTION BEFORE VS AFTER

Our analysis reveals that transformer LMs employ distinct computational strategies for filtering depending on whether the question specifying the predicate precedes or follows the collection. We briefly discussed this in Section 5 and here we provide our detailed analysis.

Table 2c shows that filter heads are minimally causal when patched from question-before to question-before prompt, even when both follow the same prompt template and item presentation style. To understand this better, we perform a multi-step causal mediation analysis.

```
Which one is a fruit in this list?
Options: Cherry, Knife, Pen, Ambulance.
\nAns:
```

```
Which one is a vehicle in this list?
Options: Binder, Peach, Watch, Scooter, Phone.
\nAns:
```

Figure 7: Example of counterfactual prompt pair used to understand the effect of patching residual latents.

Similar to the patching setup detailed in Section 2.2, we consider two prompts — $p_{\text{src}}$ and $p_{\text{dest}}$. The prompts $p_{\text{src}}$ and $p_{\text{dest}}$ have different predicates ($\psi_{\text{src}} \neq \psi_{\text{dest}}$) and sets of items ($\mathcal{C}_{\text{src}} \cap \mathcal{C}_{\text{dest}} = \emptyset$). But both prompts follow the question-before format (or both follow the question-after format). See Figure 7 for an example of the two prompts.

In the patched run $M(p_{\text{dest}})[\leftarrow h^{\ell}]$, we cache the residual stream latents at the last two tokens ($\{\backslash$"Ans", ":"$\}$) for a layer $\ell$ from the source run $M(p_{\text{src}})$ and patch them to their corresponding positions in the destination run $M(p_{\text{dest}})$. We perform this for all layers $\ell \in \{1, \ldots, L\}$ individually and track the scores (logits) of five tokens:

1. $c_{\text{src}}$ : correct answer of the source prompt, $c_{\text{src}} \in \mathcal{C}_{\text{src}} \mid \psi_{\text{src}}(c_{\text{src}})$

2. $c_{\text{dest}}$ : correct answer of the destination prompt, $c_{\text{dest}} \in \mathcal{C}_{\text{dest}} \mid \psi_{\text{dest}}(c_{\text{dest}})$

3. $c_{\text{targ}}$ : the item in the collection in the destination prompt that satisfies the predicate of the source prompt, $c_{\text{targ}} \in \mathcal{C}_{\text{dest}} \mid \psi_{\text{src}}(c_{\text{targ}})$

4. $c_{\text{oid}}$ : the item in the destination collection that shares its index with $c_{\text{src}}$ in the source collection, $c_{\text{oid}} \in \mathcal{C}_{\text{dest}} \mid \text{index}(c_{\text{oid}}, \mathcal{C}_{\text{dest}}) = \text{index}(c_{\text{src}}, \mathcal{C}_{\text{src}})$. We also make sure that $c_{\text{oid}}$ does not satisfy either predicate, $\neg\psi_{\text{src}}(c_{\text{oid}}) \wedge \neg\psi_{\text{dest}}(c_{\text{oid}})$. This token is supposed to capture if the residual states carry the positional information or order ID (Feng & Steinhardt, 2024; Prakash et al., 2025) of $c_{\text{src}}$ in the source run $M(p_{\text{src}})$.

5. $c_{\text{rand}}$ : a random item in the destination collection that does not satisfy either predicate and is different from all the other four tokens.

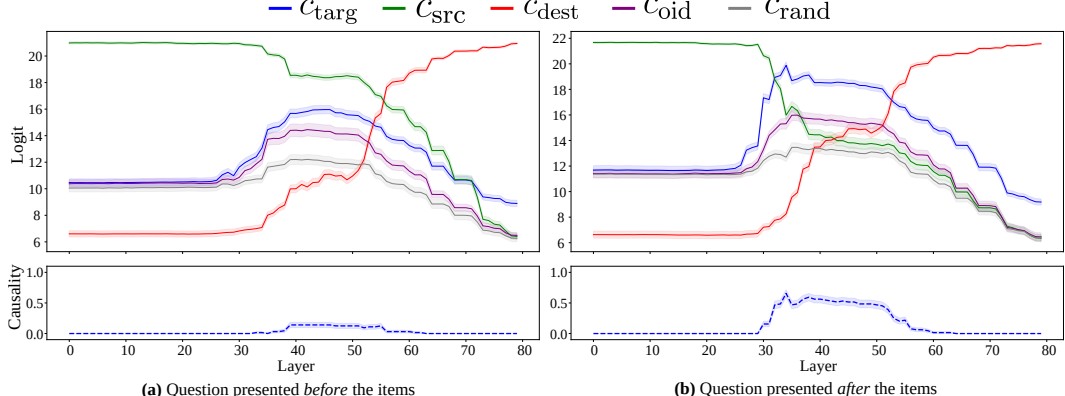

**(a)** Question presented *before* the items

**(b)** Question presented *after* the items

Figure 8: Dual implementation of filtering in LMs. In both question-before and question-after formats $c_{\text{targ}}$ (blue) shows elevated scores after patching the residual state at middle layers. But in the question-before format that score is never strong enough to dominate $c_{\text{targ}}$ (green). The violet line shows scores for $c_{\text{oid}}$, the item that shares the same index in $\mathcal{C}_{\text{dest}}$ with $c_{\text{src}}$ in $\mathcal{C}_{\text{src}}$, which also shows elevated scores in middle layers, although not as pronounced as $c_{\text{targ}}$.

We curate the source and destination prompts such that all five tokens are distinct and perform this experiment for both the question-before and question-after settings. We plot the results in Figure 8 and make the following observations.

**O1**: In both question-before and question-after settings, the score of $c_{targ}$ (blue) increases in middle layers (30-55) where we identify the filter heads to be. However in the question-before setting, that score is never strong enough to dominate $c_{dest}$ (green). While in the question-after setting $c_{targ}$ becomes the highest scoring token among the four, achieving $\sim 70\%$ causality in these critical layers.

**O2**: We notice a bump in the score of $c_{oid}$ (violet) in the middle layers, although it is not as pronounced as $c_{targ}$. This suggests that residual latents in these layers also contain the positional/order information of $c_{src}$. This has been observed in Feng & Steinhardt (2024) and Prakash et al. (2025). We also notice a slight bump in the score of $c_{rand}$ (gray).

**O3**: If the patching is performed in late enough layers ($> 60$) it copies over the final decision (red) from the source run.

The distinction between the trends of $c_{targ}$ and $c_{dest}$ in Figure 8a indicate that the LM relies more on an alternate mechanism, than the one involving the filter heads, in order to perform filtering in a prompt where the question is presented before the items. We hypothesize that the question appearing before the collection allows the LM to perform *eager evaluation*: storing an *is_match* flag for each item in the collection when they are processed. If this is true then manipulating the *is_match* flag should cause predictable changes in the LM's behavior in the question-first setting, while the question-after setting should not be sensitive to such manipulations.

**Effect of ablating the *is_match* flag.** If the LM is indeed using the *is_match* flag to perform filtering in question-before settings, we would expect that ablating this flag would significantly degrade performance.

To test this hypothesis, we ablate the *is_match* flag in an item by replacing all of the residual stream latents of the tokens of that item with their corresponding latents cached for the same item in a neutral prompt (see Figure 9 for details). When we perform this ablation for each of the items in the collection, we indeed see a significant drop in LM performance for the question-before setting, while the performance remains mostly unchanged for the question-after setting (see Table 8).

This experiment supports our hypothesis that the question-before setting allows the LM to *eagerly* evaluate whether an item satisfies the predicate or not, store this in-

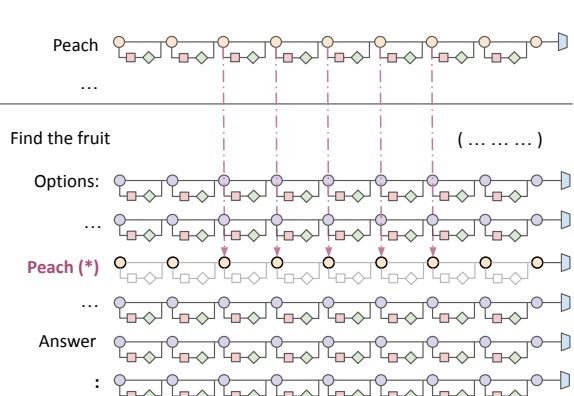

Figure 9: Ablating the *is_match* flag. For an item (e.g "Peach") in the collection, we cache the residual stream latents corresponding to the tokens of that item from a neutral prompt (e.g. any text w/o any predicate that contains the word "Peach"). Then in a separate pass, we replace the latents corresponding to that item in the original prompt with the cached latents. This effectively removes any information about whether that item satisfies the predicate or not.

termediate result in the residual latents of the items as it processes the collection, and relies on that to make the final decision. We also notice that the question-after setting shows minimal sensitivity to this flag-ablation, which suggests that the processing of items do not rely on the context here: the LM populates the semantics (*enrichment* in Geva et al. (2023)) of each item in a context independent manner first, and then applies the predicate to perform filtering when the question is presented after.

**Effect of swapping the *is_match* flag between two items.** Our most decisive evidence comes from *swapping* the *is_match* flag between items: if we swap the *is_match* flag stored in $c_{pos}$ that satisfies the predicate with $c_{neg}$ that does not satisfy the predicate, we should expect the LM to

Table 8: Effect of ablating *is_match*. Evaluated on 512 examples from the *SelectOne* task.

| Ques Place | W/o *is_match* Acc |
|---|---|
| Before | 46.09% |
| After | 96.06% |

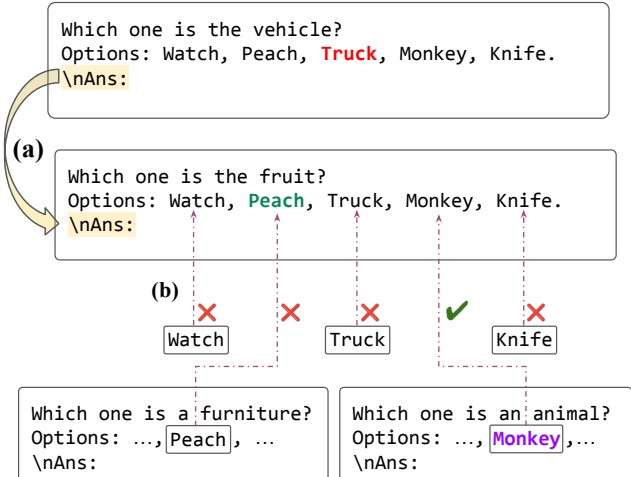

Figure 10: Counterfactual patching setup to swap the *is_match* flag. We perform a two-part intervention to determine whether the model stores filtering decisions as flags in item representations. **(a)** We patch the residual states at the final two token positions ("\nAns" and ":") from a source prompt $p_{\text{src}}$ to a destination prompt $p_{\text{dest}}$ at a single layer. **(b)** Additionally, for an item $c \in \mathcal{C}_{\text{dest}}$, we replace its hidden representations across *all* layers with those from a prompt $p_{\text{diff}}$ containing a different predicate $\psi_{\text{diff}} \notin \{\psi_{\text{src}}, \psi_{\text{dest}}\}$. We use different $p_{\text{diff}}$ (with different $\psi_{\text{diff}}$) per item in $\mathcal{C}_{\text{dest}}$. Crucially, we ensure exactly one item $c_{\text{flag}}$ (distinct from both the source and destination answers) carries the *is_match* flag from its corresponding $p_{\text{diff}}$.

change its answer from $c_{\text{pos}}$ to $c_{\text{neg}}$ if it relies on the *is_match* flag. To test this hypothesis, we set up another activation patching experiment (illustrated in Figure 10, which is a more elaborate version of Figure 5 in the main text). We perform a 2 part intervention:

**I1**: Similar to Figure 7, we consider two prompts $p_{\text{src}}$ and $p_{\text{dest}}$ that follow the same format, either question-before or question-after, but with different predicates, $\psi_{\text{src}} \neq \psi_{\text{dest}}$. However, now they operate on the same collection of items, $\mathcal{C}_{\text{src}} = \mathcal{C}_{\text{dest}} = \mathcal{C}$. We perform the same intervention, patching the residual stream latents at the last two token positions from $M(p_{\text{src}})$ to $M(p_{\text{dest}})$ for a layer $\ell$. And we track the scores of $c_{\text{src}}, c_{\text{dest}}, c_{\text{rand}}$ as defined before. Notice that as the collections are the same, $c_{\text{targ}} = c_{\text{oid}} = c_{\text{src}}$.

**I2**: In addition, we choose another item $c_{\text{flag}} \in \mathcal{C}$ that is different from $c_{\text{src}}, c_{\text{dest}}, c_{\text{rand}}$ and perform the following intervention to make sure that only $c_{\text{flag}}$ carries the *is_match* flag while none of the other items do. In order to achieve that we cache $c_{\text{flag}}$'s latents from an alternate prompt $p_{\text{flag}}$ with a predicate $\psi_{\text{flag}}$ which is satisfied by $c_{\text{flag}}, \psi_{\text{flag}}(c_{\text{flag}})$. Then in the patched run, we replace the latents corresponding to $c_{\text{flag}}$ in $M(p_{\text{dest}})$ with the cached latents from $M(p_{\text{flag}})$. This makes sure that $c_{\text{flag}}$ now carries the *is_match* flag.

Similarly, to make sure that an item $c' \in \mathcal{C} \backslash \{c_{\text{flag}}\}$ does not carry the *is_match* flag, we cache its latents from another example $p'$ with a predicate $\psi'$ such that $\neg \psi'(c')$. Then we replace the latents corresponding to $c'$ in $M(p_{\text{dest}})$ with the cached latents from $M(p')$. We perform this for all items in $\mathcal{C} \backslash \{c_{\text{flag}}\}$. This effectively ensures that only $c_{\text{flag}}$ carries the *is_match* flag while all other items do not. See Figure 10 for an illustration.

In Figure 11 we plot the results of this experiment for both question-before and question-after settings. For a layer $\ell$, **I1** is applied for only that layer, without or with **I2**, which is applied to *all* layers.

As expected, we see that in the question-after setting applying **I2** with **I1** is almost indistinguishable from just applying **I1**. **I2** has minimal effect because the LM cannot rely on the *is_match* flag when the question comes after.

However, in the question-before setting, we observe that the score trend of $c_{\text{flag}}$ (violet) and $c_{\text{dest}}$ (green) almost swap their positions in only **I1** versus when **I2** is applied in addition. With the flag-swap intervention **I2**, the LM systematically picks $c_{\text{flag}}$ as the answer in the early layers. This further validates our hypothesis that the LM is relying on the *is_match* flag to make the final decision in the question-before setting.

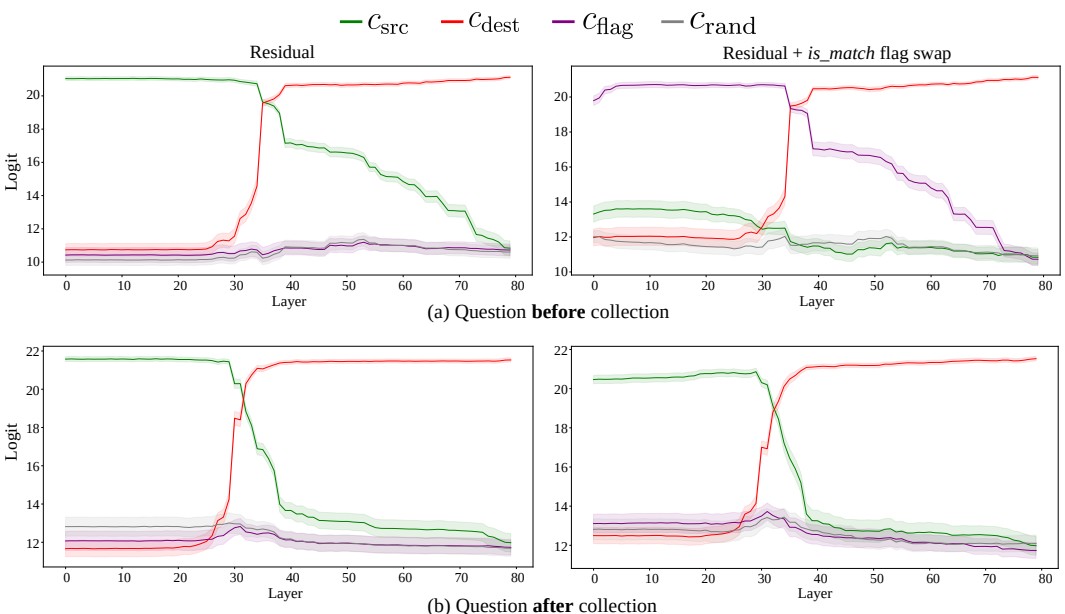

Figure 11: Effect of swapping the *is_match* flag between items. The figures on the left shows the effect of patching only the residual states (Figure 7) and figures on the right are when we additionally swap the *is_match* with another item (Figure 10). The pair of figures on the top shows both cases in the question-before format **(a)**. We observe that $c_{\text{flag}}$ becomes the top scoring item when the patching is performed in early layers. But this swapping of flags has no effect in the question after case, pair of figures on the bottom **(b)**.

We do not claim that the LM *only* relies on the *is_match* flag in the question-before setting. The fact that we see a bump in the score of $c_{\text{targ}}$ (blue) in Figure 8 and patching $q_{\text{src}}$ from question-before to question-after prompts has non-trivial causality (see Table 2c) indicates that the LM does carry the predicate information in middle layers even in the question-before setting, although not as strongly as in the question-after setting.

This dual filtering implementation strategy — lazy evaluation via filter heads versus storing eager evaluation information with *is_match* flag — exemplifies a broader principle in neural computation: transformer LMs can maintain multiple pathways for core operations, dynamically selecting strategies based on what information is available. And the fact that filter heads still remain partially active even in the question-first setting shows that these mechanisms operate in parallel rather than in mutual exclusion.

## C  DIFFERENT APPROACHES FOR LOCATING THE FILTER HEADS

In this section we discuss the different approaches we explored to identify the filter heads.

**Filter Score.**  To capture the filtering behavior of the heads based on their attention pattern, we design a *filter score* that quantifies the extent to which a head focuses its attention on the elements satisfying the predicate $\psi$ over other elements in $\mathcal{C}$.

$$\text{FilterScore}([\ell, j], \mathcal{C}, \psi) = \text{score}_{\ell j}(c \mid \psi(c)) - \max_{\neg\psi(c)} \left(\text{score}_{\ell j}(c)\right) \quad (6)$$

$$\text{where,} \quad \text{score}_{\ell j}(c) = \sum_{t \in c} \text{Attn}_{[\ell,j]}(q_{-1}, K)_t$$

While calculating FilterScore we make sure that there is only one item $c \in \mathcal{C}$ such that $\psi(c)$ is true. The FilterScore then select heads based on how much they focus their attention on the correct item over the most attended incorrect item. The score function sums up the attention scores over all tokens in an item $c$ to account for multi-token items. Note that the score is calculated based on the attention pattern at the last token of the prompt ("`:`").

We notice that heads in a range of middle layers exhibit stronger filtering behavior compared to those in the earlier or later layers.

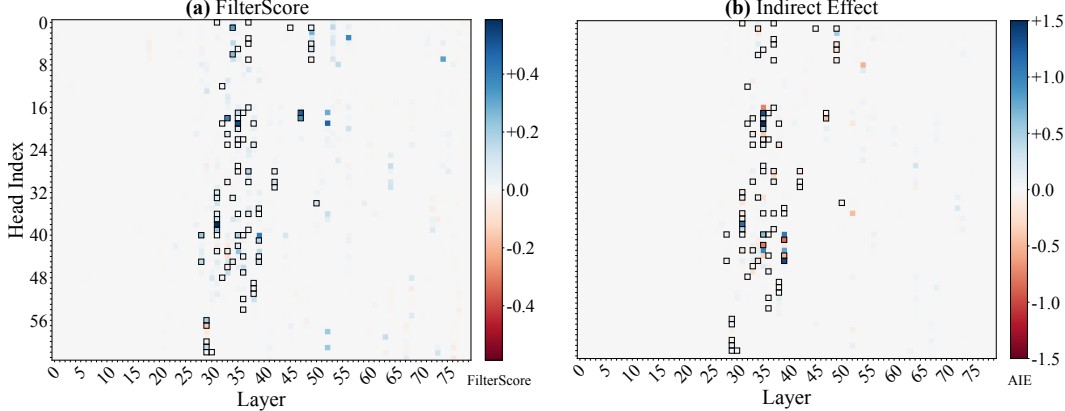

Figure 13: Location of filter heads in Llama-70B. **(a)** shows the individual FilterScore for each head: how much they attend to the correct option over others. **(b)** shows the indirect effect: how much patching $q_{src}$ from a single head promotes the predicate target. The filter heads identified with Section 2.3 are marked with black borders.

**Activation Patching.** We can patch the attention heads *individually* and quantify their *indirect effect* at mediating the target property in the patched run. Todd et al. (2024) and Feucht et al. (2025) identified Function Vector heads and Concept heads with this approach. Specifically, for each head $[\ell, j]$ we patch $q_{src}$ from the source run to the destination run with the method detailed in Section 2 and check its effect on boosting the score (logit) of the target item, $c_{targ}$. The indirect effect is measured as $logit[\leftarrow q_{src}](c_{targ}) - logit(c_{targ})$.

**Activation Patching with Learned Masks.** In our analysis we find that patching a single head is not enough to causally influence the LLM behavior. This is because the same mechanism can have parallel implementations (McGrath et al., 2023), which may work against the intervention to rectify its effects. To overcome this issue, we learn a differential mask to locate the minimal set of components (attention heads) that collectively implement filtering. Following De Cao et al. (2020);

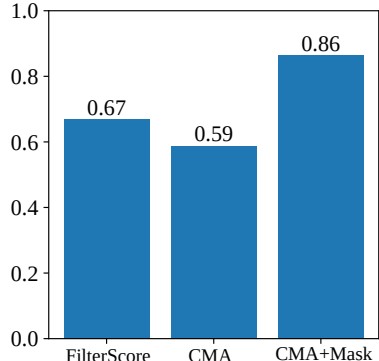

Figure 12: Collective causality of 79 filter heads identified with FilterScore, CMA, and CMA with learned masks. Evaluated on the same 512 examples from the *SelectOne* task

Davies et al. (2023); and Csordás et al. (2021), we optimize a differential binary mask over all of the attention heads. The idea is to learn a binary mask for the neural components (query states of attention heads in our case) such that after an interchange intervention (Eq. (3)) is performed, the LM shows the desired counterfactual behavior (executing the predicate from the source prompt).

When we compare different head identification approaches while controlling for the number of heads selected, the learned mask approach achieves the highest causality (Figure 12).

## D  VECTOR ALGEBRA WITH PREDICATE REPRESENTATION

We explore the geometric properties of the predicate representation $q_{src}$ by examining its behavior under vector arithmetic operations. Specifically, we investigate whether, when we compose two predicates (*find the fruit* and *find the vehicle*) by adding their corresponding $q_{src}$ vectors, this resulting vector represents a meaningful combination of the two predicates.

**Adding predicate representations results in disjunction of the predicates.** If we add the $q_{src}$ vectors of two source prompts with different predicates, $p_{src1} = \mathbb{P}(\mathcal{C}_1, \psi_1 = \textit{is\_fruit})$ and $p_{src2} = \mathbb{P}(\mathcal{C}_2, \psi_2 = \textit{is\_vehicle})$, we find that the resulting vector $q_{composed} = q_{src1} + q_{src2}$ can be used on a destination prompt $p_{dest} = \mathbb{P}(\mathcal{C}_3, \psi_3 \notin \{\textit{is\_fruit}, \textit{is\_vehicle}\})$ to execute the disjunction of the two predicates (i.e., *find the fruit or vehicle*) in $\mathcal{C}_3$. The setup is illustrated in Figure 14 with an example from the *SelectOne* task.

We conduct this experiment for the *SelectOne* task. We compose $q_{composed}$ with two prompts and curate $p_{dest}$ such that there is only one item $c_{targ}$ in $\mathcal{C}_{dest}$ that satisfies the composed predicate $\psi_{src1} \cup \psi_{src2}$. We patch $q_{composed}$ to the destination run and consider the intervention to be successful

if the LM thinks $c_{\text{targ}}$ is the most probable item. We achieve a causality score of $0.6523$ (334 out of 512) with logit of $c_{\text{targ}}$ increased by $6.75 \pm 3.94$ after the intervention.

This shows that the predicate representations are compositional, allowing for the construction of more complex predicates through simple vector operations.

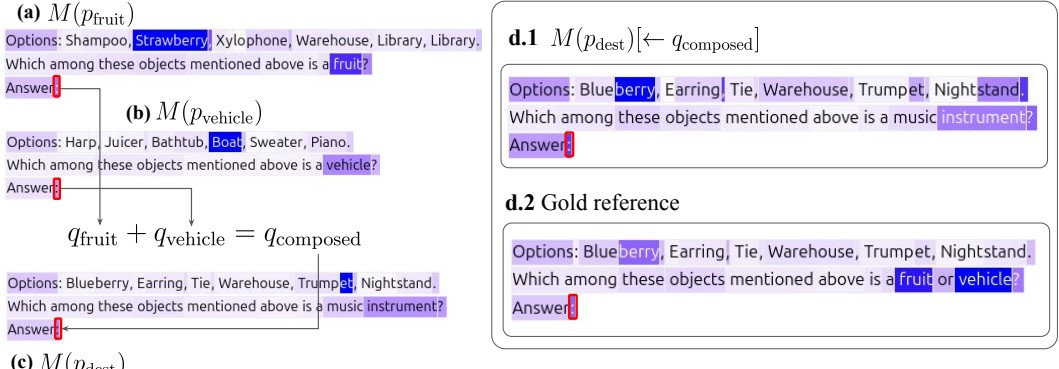

Figure 14: Aggregated attention pattern of the filter heads from the last token position, with the composition setup. The predicate encoding $q_{\text{src}}$ is collected from 2 prompts, **(a)** $p_{\text{src1}} = \mathbb{P}(\mathcal{C}_1, \psi_1 = \texttt{is\_fruit})$ and **(b)** $p_{\text{src2}} = \mathbb{P}(\mathcal{C}_2, \psi_2 = \texttt{is\_vehicle})$. Their addition $q_{\text{composed}}$ is patched to the destination run. The resulting attention pattern shown in **(d.1)** indicates that now filter heads select the items in $p_{\text{dest}}$ that satisfy $\psi_{\text{src1}} \cup \psi_{\text{src2}}$. **(d.2)** shows the attention pattern for a gold prompt with the disjunction predicate.

## E    DISTINGUISHING ACTIVE FILTERING FROM ANSWER RETRIEVAL

Our identification of filter heads raises two critical questions about their computational role. First, do these heads actively perform filtering, or do they merely attend to items that were already filtered by earlier layers? Second, do they encode the abstract predicate (e.g., "is a fruit") or simply match specific answers from context? To establish that filter heads actively perform filtering rather than passively attending to pre-filtered results, we designed causal intervention experiments where query states carrying predicate information are transferred between prompts. The consistent ability of these transferred queries to trigger the transported filtering operation on entirely different collections demonstrates that the heads actively apply predicates rather than simply reading pre-computed results.

### E.1    DO THESE HEADS MERELY MATCH WITH A PRE-COMPUTED ANSWER?

A more subtle concern is whether these heads encode abstract predicates or simply store concrete answers. For instance, when the source prompt asks to *find the fruit* with the answer being *Plum*, does the query state encode the predicate *find the fruit* or the specific item *Plum*? In the patched run, the *Plum* representation would naturally show higher similarity to the representation of *Apple* (another fruit) than to *Watch* (a non-fruit), potentially explaining the selective behavior we observe in the attention pattern after patching $q_{\text{src}}$ in the patched run.

To resolve this ambiguity, we designed a critical experiment: we use source prompts that contain predicates but no valid answers. We observe that even in such cases, the filter heads retain their high causality of $0.80$ (410 out of 512 examples from the *SelectOne* task, $\Delta c_{\text{targ}} = 8.08 \pm 3.1$). Combined with our ablation studies in Section 3.3, these experiments demonstrate the crucial role of filter heads in actively participating in filtering, rather than simply mediating the pre-filtered items.

## F    AVERAGING TO REMOVE THE ORDER ID

While in Appendix E we make the case that filter heads encode predicates rather than specific answers, our error analysis reveals that sometimes filter heads additionally transfer the position of the answer in the source prompt.

When we examine the cases where our intervention fails, we find that the model sometimes selects the item at the same position as the original answer in the source prompt. Figure 8 also shows elevated scores for items that match the position of the source answer $c_{\text{oid}}$ in critical layers, although not as

high as $c_{\text{targ}}$. This suggests that the LM also encodes the positional information or *order ID* (Feng & Steinhardt, 2024; Prakash et al., 2025) of $c_{\text{src}}$ alongside the predicate. A probable explanation for this is that, since these filter heads are distributed across a range of layers, patching $q_{\text{src}}$ from filter heads in later layers may bring over specific contributions from filter heads in earlier layers.

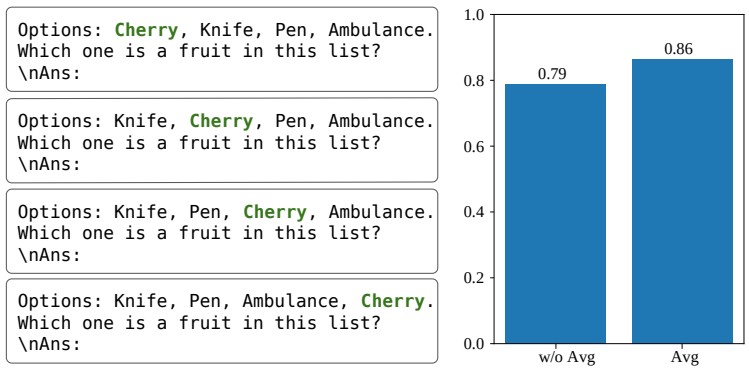

Figure 15: Averaging $q_{\text{src}}$ to remove the order ID information. This simple trick improves the causality scores by 7 to 10 points across the board. Causality scores presented for 512 examples from the *SelectOne* task.

To isolate the predicate signal from this positional bias, we use a simple trick: averaging the query states across multiple source prompts. We produce $n$ variations of the same source prompt $p_{\text{src}} = \mathbb{P}(\mathcal{C}_{\text{src}}, \psi_{\text{src}})$ by changing the index of the correct answer $c_{\text{src}}$. Figure 15 illustrates this idea. This simple averaging improves our causality scores.

## G  ADDING A PRIMING PREFIX HELPS WITH CAUSALITY.

Table 1 shows that filter head causality varies across information types for the same *SelectOne* task. Notably, tasks requiring recalling a person's nationality and the location of a landmark show lower causality scores.

Following Amini & Ciaramita (2023), we provide contextual priming and check the causality in these cases. In a question-after format, if before presenting the items we add a prefix that explicitly instructs the LM to recall relevant information, we can achieve approximately a 10-point improvement in causality scores. This experiment further validates the hypothesis that filter heads work better when the relevant semantic information required for filtering is already present in the item latents.

Table 9: Priming the context helps improve the causality score.

| Filtering Task | W/O Priming | With Priming | Priming Prefix |
|---|---|---|---|
| Person Nationality | 0.504 (258/512) | 0.625 (320/512) | `Recall the nationality of these people:\n` |
| Landmark in Country | 0.576 (295/512) | 0.670 (343/512) | `Recall which country these landmarks are located in:\n` |

# H ABLATION ON DESIGN CHOICES

In this section we conduct ablation studies on two design choices we made in this work.

**Sparsity Regularizer.**    To identify the minimal set of heads necessary for filtering, we incorporate the L1 norm of the mask in our optimization objective. This sparsity constraint ensures we select only the most influential heads while filtering out noisy ones.

$$\mathcal{L} = \underbrace{-\text{logit}(c_{\text{targ}}) + \frac{1}{|\mathcal{C}_{\text{dest}}| - 1} \sum_{\substack{c \in \mathcal{C}_{\text{dest}} \\ c \neq c_{\text{targ}}}} \text{logit}(c)}_{\text{Target Loss}} + \underbrace{\lambda ||\text{mask}||_1}_{\text{Sparsity}} \tag{7}$$

The target loss maximizes the logit of $c_{\text{targ}}$ (the item satisfying the source predicate $\psi_{\text{src}}$), while suppressing the other items in $\mathcal{C}_{\text{dest}}$. The sparsity penalty, modulated by the hyperparameter $\lambda$, pushes the mask towards zero so that fewer heads are selected.

We find that selecting the right sparsity coefficient is crucial to achieve higher causality. Through a systematic search across $\lambda \in (0, 1]$, while controlling for other variables (training data, other hyperparameters) of the optimization process, we find a clear sweet spot at $\lambda = 0.02$. In Figure 16 we report the causality of identified heads across format, where the source prompt is from *SelectOne* and the destination prompt is from *SelectOne-MCQ*. While evaluating, we do not employ the averaging trick (Appendix F), and transfer head states at only the last token.

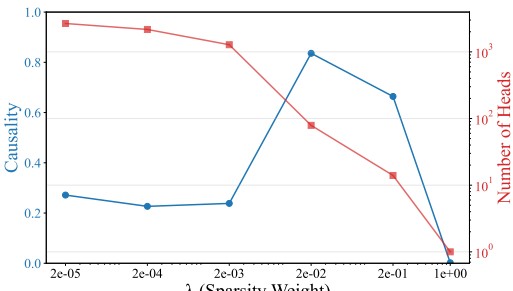

Figure 16: Sparsity-performance tradeoff. Optimal causality achieved with 79 heads identified with $\lambda = 0.02$. Lower values of $\lambda$ includes noisy heads while larger values over-prune critical heads.

This tradeoff is intuitive: insufficient regularization allows noisy irrelevant heads into the mask, reducing generalization performance. And excessive regularization ends up pruning essential heads, degrading causality. We have identified this optimal value of $\lambda$ with Llama-70B but use the same value for all the models we investigate in this paper.

**Last two tokens for evaluation.**    Autoregressive transformer LMs process texts sequentially without knowing what comes next. When the LM reaches the second to last token position "\nAnswer", it doesn't know that ":" will follow. So, it may begin performing filtering at "\nAnswer", which will be accessible from the ":" token. This creates a potential problem for our intervention: if we only perform our intervention at ":", pre-computed values "\nAnswer" may *leak* at or after our intervention location, reducing the efficacy.

To partially address this issue, during evaluation we patch the query states of filter heads for last two tokens, both "\nAnswer" and ":". In our controlled setup, we ensure that our source and destination prompts always share these same last two tokens, which allows us to do this intervention.

Table 10: Evaluation by patching only the last token for *SelectOne* task.

| Semantic Type | Causality | $\Delta$logit |
|---|---|---|
| Object | 0.819 (839/1024) | $+8.12 \pm 2.93$ |
| Person Profession | 0.842 (431/512) | $+7.31 \pm 3.04$ |
| Person Nationality (+ Priming Prefix) | 0.629 (322/512) | $+6.28 \pm 3.75$ |
| Landmark in country (+ Priming Prefix) | 0.678 (347/512) | $+7.05 \pm 3.20$ |

For the *SelectOne-Object* task, single-token patching achieves a causality of 0.819 — only slightly lower than the 0.863 we report in the main text with two-token patching (Table 1). This suggests that while preventing information leakage helps, it is not as critical as we initially anticipated. The filter heads remain robustly causal even with this simpler intervention.

# I    DECOMPOSING $W_{QK}$ INTO SMALLER FUNCTIONAL UNITS WITH SVD

We investigate whether attention heads are too large of a model unit for analyzing the filtering task and if we can be more surgical by decomposing heads into smaller functional units. We take inspiration from Merullo et al. (2024b) and attempt to locate low-rank subspaces of the $QK$ matrix necessary to perform the filtering task.

**Approach.**    In attention heads, the query projection $W_Q$ and the key projection $W_K$ *read* from the residual stream latent $h_t$ to determine the attention pattern, while the $OV$ projection $W_{OV}$ *writes* back to the residual stream. Equation (1) shows how the attention distribution from a query token $t$ is calculated, but for simplicity it hides some technical details, such as the fact that a pre-layernorm is applied on $h^{\ell-1}$ before $W_K$ and $W_Q$ are applied, and RoPE positional encoding (Su et al., 2024) is applied on the resulting query state and key states before the attention score is calculated. If we do not ignore these details the pre-softmax attention score from the $q^{th}$ token to the $k^{th}$ token is calculated as

$$a_{qk} = \text{RoPE}(\tilde{h}_q W_Q) \left(\text{RoPE}(\tilde{h}_k W_K)\right)^T, \quad \text{where } \tilde{h} = \text{LayerNorm}(\text{h}) \tag{8}$$

When we ignore the positional encoding (i.e. setting $\text{RoPE} = I$), we can combine $W_Q$ and $W_K$ projections in to a single bilinear form.

$$a_{qk} = \tilde{h}_q W_Q \left(\tilde{h}_k W_K\right)^T = \tilde{h}_q W_Q W_K^T \tilde{h}_k^T = \tilde{h}_q W_{QK} \tilde{h}_k^T \tag{9}$$

Note that although $W_{QK}$ is a $d_{\text{embd}} \times d_{\text{embd}}$ matrix it has a rank of $d_{\text{head}}$. This is because $W_Q$ and $W_K$ both have a rank of $d_{\text{head}}$. To get the attention distribution from the $q^{th}$ token the attention score $a_{qk}$ is calculated for all $k \leq q$, which is then divided by a scaling factor $\sqrt{d_{\text{head}}}$ before applying softmax (see Equation (1)).

If we take the SVD of $W_{QK}$, we can rewrite it as a sum of rank-1 matrices, the outer products of the left and right singular vectors, scaled by their corresponding singular values.

$$W_{QK} = U\Sigma V^T = \sum_{i=0}^{d_{\text{head}}} \sigma_i * u_i v_i^T \tag{10}$$

Merullo et al. (2024b) found that a subset of these rank-1 components of $W_{QK}$ read from low-rank subspaces of the residual stream, which has been written into it by specific components of $W_{OV}$ of the heads from earlier layers, forming a low-rank communication channel via the residual stream. In this section we investigate if filtering is implemented by the specific components of $W_{QK}$.

For a head $[\ell, j]$ we want to learn a binary mask $\in \{0,1\}^{d_{\text{head}}}$ that selects specific components of $W_{QK}$ crucial for the filtering task. In the patched run we recalculate $a_{qk}$ by selectively patching these components:

$$a_{qk} = u_{\text{patch}} \Sigma V^T \tilde{h}_k^T \tag{11}$$
$$\text{Where, } u_{\text{patch}} = u_{\text{dest}} + (u_{\text{src}} - u_{\text{dest}}) \odot \text{mask}$$
$$u_{\text{src}} = \tilde{h}_{\text{src}} U \ , \ \text{the left projected latent from source } p_{\text{src}}$$
$$u_{\text{dest}} = \tilde{h}_{\text{dest}} U \ , \ \text{the left projected latent from destination } p_{\text{dest}}$$
$$\text{and, } \odot \text{ denotes element-wise multiplication} \tag{12}$$

Note that $u_{\text{patch}}$ is a $d_{\text{head}}$ vector where the $i^{th}$ value is the same as the $i^{th}$ value of $u_{\text{src}}$ when $\text{mask}_i$ is on, and the same as the $i^{th}$ value of $u_{\text{dest}}$ when $\text{mask}_i$ is off. Recalculating $a_{qk}$ in the patched run ensures that only the information encoded in the selected subspace is transferred from the source prompt while the information in other components is preserved. Similarly to the method detailed in Section 2, we learn this mask for all heads.

Figure 17 shows that middle-layer attention heads contain more components relevant to the filtering task, which is consistent with where we locate filter heads. Interestingly, while our optimization

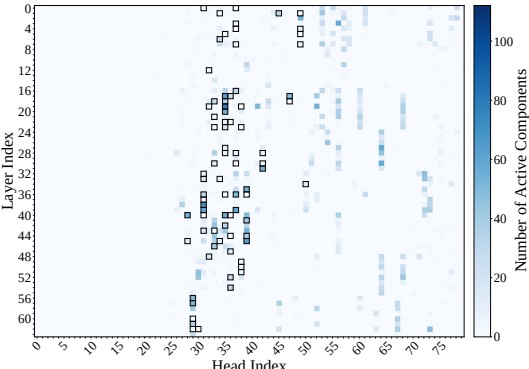

Figure 17: Distribution of critical SVD components of $W_{QK}$ across attention heads in Llama-70B. Middle layers show higher component counts, which aligns with filter head locations.

Table 11: Causality of the SVD components (without the averaging trick) on the *SelectOne* task. Components located using object-type selection (e.g. *find the fruit*) generalize to person identification (e.g. *find the actor*). Both in-distribution and out-of-distribution causality were calculated using 512 examples. Excluding the components in later layers significantly improves performance in both the in-distribution and out-of-distribution validation sets.

| Attention Heads | Object-type | Profession |
|---|---|---|
| All | 0.627 | 0.420 |
| Heads in $\ell < 52$ | 0.811 | 0.773 |
| Only filter heads | 0.766 | 0.752 |

identifies components in later layers, excluding these actually improves causality (Table 11), which suggests that they may introduce noise rather than information relevant for filtering. Most notably, using only the SVD components of previously identified filter heads achieves high causality (0.766 for in-distribution, 0.752 for out-of-distribution), demonstrating that predicate information is concentrated in low-rank subspaces within specialized heads. This finding indicates that filter heads implement filtering through coordinated interactions between low-rank subspaces of the query and key states, rather than using their full representational capacity.

## J  WHAT DO THE FILTER HEADS WRITE TO THE RESIDUAL STREAM?

Our previous experiments established that the filter heads encode the filtering criteria (the *predicate*) in their query states and perform filtering through query-key interactions. But what information do these heads actually *write* back to the residual stream? We investigate two possibilities:

**H1**: Filter heads write the filtered item directly. For example, writing *apple* when filtering for fruits or *Tom Cruise* when filtering for actors.

**H2**: Filter heads write the positional information or the *order id* of the filtered items (e.g., *the second item*). And this information is then dereferenced based on the task/format requirements with a *lookback* mechanism identified in Prakash et al. (2025).

To distinguish between these hypotheses, we design a controlled causal mediation analysis by patching the output of the filter heads (output of the OV projection). We create source and destination prompts with different predicates but the same collection of items presented in different orders (see Figure 18 left). This ensures that $c_{\text{src}}$, the correct answer of the source prompt remains a valid option in both prompts, enabling us to test **H1**. While shuffling, we make sure that the order of $c_{\text{src}}$ is different so that we can test **H2**. We choose an MCQ format and ensure that the MCQ labels differ between the prompts to avoid confounding effects of this intervention.

We patch the output (OV contribution to the residual stream) of 79 filter heads identified with the *SelectOne* task (not MCQ). The results plotted in Figure 18 (right) reveal nuanced insights:

1. The logit of $c_{\text{src}}$, the answer in the source prompt, and the target of the transferred predicate, increase significantly after patching. The MCQ label associated with $c_{\text{src}}$ in the destination prompt also elevates significantly. This provides some evidence in support of **H1** that filter heads do write filtered items directly to the residual stream. However, we acknowledge that these elevated scores may also be explained if filter heads in earlier layers also mediate the predicate information, which is then read and executed by the filter heads in later layers.

2. The logit of $c_{\text{oid}}$ and its MCQ label also increase, suggesting that filter heads write the positional information as well (support for **H2**).

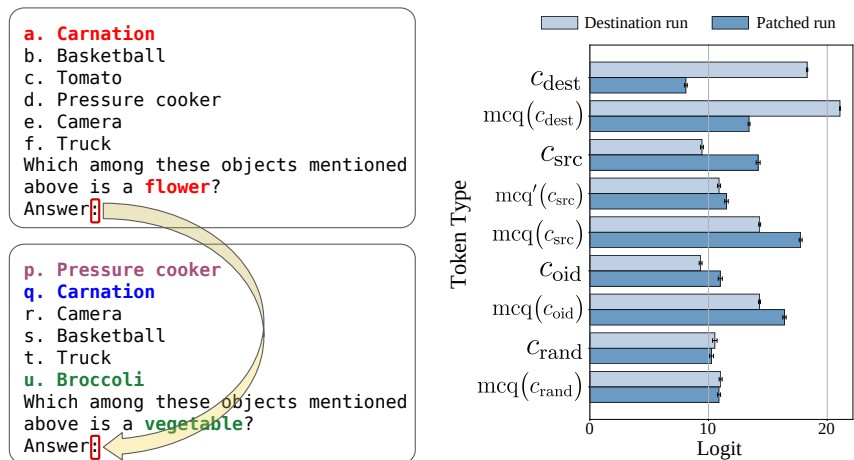

Figure 18: Testing what filter heads write to the residual stream. **Left**: Source and destination prompts contain identical items in different orders with different MCQ labels. **Right**: Logit changes after patching the filter head outputs. $\mathrm{mcq}(c)$ denotes the MCQ label of the item $c$ in the destination prompt and $\mathrm{mcq}'(c)$ denotes the MCQ label in the source prompt. Both $c_{\mathrm{src}}$ and $c_{\mathrm{oid}}$ (and the MCQ labels in destination prompt) show increased scores, suggesting that filter heads write both the filtered content and its positional information to the residual.

Although our findings from this patching experiment are somewhat inconclusive, they suggest that filter heads write both *what* items are filtered and *where* they are located. This dual encoding strategy aligns with our earlier finding of multiple implementation pathways. This redundancy may enable the LM to flexibly use either piece of information, or both of them in parallel.

## K  DISTINGUISHING FILTER HEADS FROM OTHER SPECIALIZED ATTN HEADS

We compare filter heads with two previously characterized attention head types to establish their distinctiveness. Function Vector (FV) heads, identified in Todd et al. (2024), encode a compact representation of the task demonstrated in few-shot examples. And, Concept Induction (CI) heads, identified in Feucht et al. (2025), encode the concept or lexical unit in multi-token words.

Following the identification procedures detailed in the respective papers, we locate FV and CI heads in Llama-70B. We use publicly available dataset and source code from Todd et al. (2024) and Feucht et al. (2025). In Figure 19 we plot the distribution of these heads with filter heads identified with *SelectOne* task. Interestingly, although both FV heads and filter heads are concentrated in the middle layers, only 7 (out of 79) heads overlap. This separation suggests that these head types play parallel, complementary roles in LLM computation.

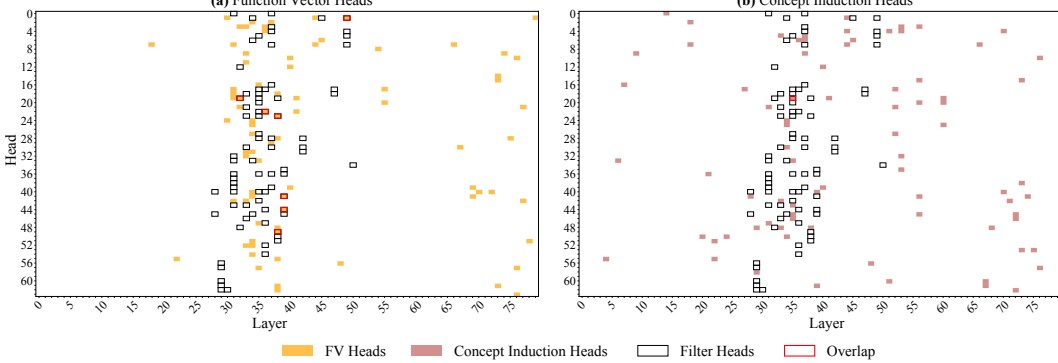

Figure 19: Distribution of specialized attention heads in Llama-70B. Both Filter and FV heads are concentrated in the middle layers, but there are only 7 overlapping heads. CI heads exhibit weaker concentration in the middle layers, with minimal overlap with filter heads (only one head). For a fairer comparison, we choose 79 heads from each head type.

To verify whether these head types serve different computational roles, we check their *causality* and perform ablation studies. The near-zero causality for FV and CI heads confirms that they don't directly contribute to filtering. Ablating filter heads catastrophically degrades the task performance, while removing other head types has a limited impact. The 12% performance drop when ablating FV heads, despite their minimal causality score, suggests that they contribute indirectly — possibly encoding some task-specific information such as the format of the expected answer.

Table 12: Causality and necessity of different head types in filtering task. Evaluated on the same 512 example pairs where the LM achieves 100% accuracy without any ablation. Filter heads show strong *causality* and ablating them significantly reduces the task performance. Both FV and CI heads show minimal causality. However, ablating FV heads reduces the task performance by 12%.

| Head-type | Causality | LM Acc after ablation |
|---|---|---|
| Filter | 0.863 | 22.5% |
| Function Vector | 0.002 | 88.3% |
| Concept Induction | 0.080 | 98.2% |
| Random | 0.00 | 99.6% |

The clean separation between FV heads and filter heads, despite being concentrated in similar layers, suggests that transformer LMs have developed distinct, specialized modules that operate in parallel.

## L    REPLICATING EXPERIMENTS IN GEMMA-27B

We have replicated our core experiments for Gemma-27B and we observe that the results mostly align with our findings in Llama-70B.

### L.1    LOCATION OF THE FILTER HEADS

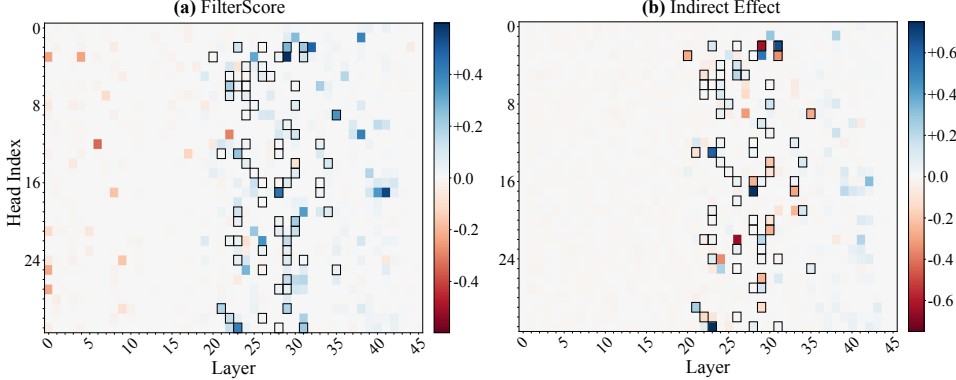

Figure 20: Location of filter heads in Gemma-27B-it. Similar to Figure 13, **(a)** shows the individual FilterScore for each head: how much they attend to the correct option over others. **(b)** shows the indirect effect: how much patching $q_{src}$ from a single head promotes the predicate target. The filter heads identified with the method detailed in Section 2.3 are marked with black borders.

### L.2    WITHIN TASK GENERALIZABILITY

Table 13: Causality of filter heads on *SelectOne* tasks. Heads identified using object-type filtering generalize to semantically distinct predicates like profession identification. Compare with Llama-70B scores in Table 1.

| Semantic Type | Causality | $\Delta$logit | With Priming |
|---|---|---|---|
| Object Type | 0.824 | +9.95 | - |
| Person Profession | 0.770 | +9.10 | - |
| Person Nationality | 0.305 | +5.98 | 0.404 |
| Landmark in Country | 0.410 | +6.48 | 0.455 |
| Word rhymes with | 0.018 | +0.12 | 0.037 |

Table 14: Portability of predicate representations in Gemma-27B filter heads across linguistic variations. Compare with Table 2. The predicate vector $q_{\text{src}}$ is extracted from a source prompt and patched to destination prompts in **(a)** different languages, **(b)** different presentation formats for the items, and **(c)** placing the question before or after presenting the collection.

| | To | | | | |
|---|---|---|---|---|---|
| **From** | English | Spanish | French | Hindi | Thai |
| English | **0.824** | 0.822 | 0.777 | 0.840 | 0.852 |
| Spanish | 0.822 | **0.815** | 0.834 | 0.820 | 0.818 |
| French | 0.760 | 0.811 | **0.779** | 0.805 | 0.807 |
| Hindi | 0.787 | 0.816 | 0.834 | **0.822** | 0.838 |
| Thai | 0.805 | 0.816 | 0.807 | 0.793 | **0.797** |

**(a)** Cross-lingual transfer

| | To | |
|---|---|---|
| **From** | single line | bulleted |
| single line | **0.824** | 0.805 |
| bulletted | 0.826 | **0.799** |

**(b)** Across option presentation style

| | To | |
|---|---|---|
| **From** | after | before |
| after | **0.824** | 0.412 |
| before | 0.190 | **0.055** |

**(c)** Placement of the question

## L.3 ACROSS TASK GENERALIZABILITY

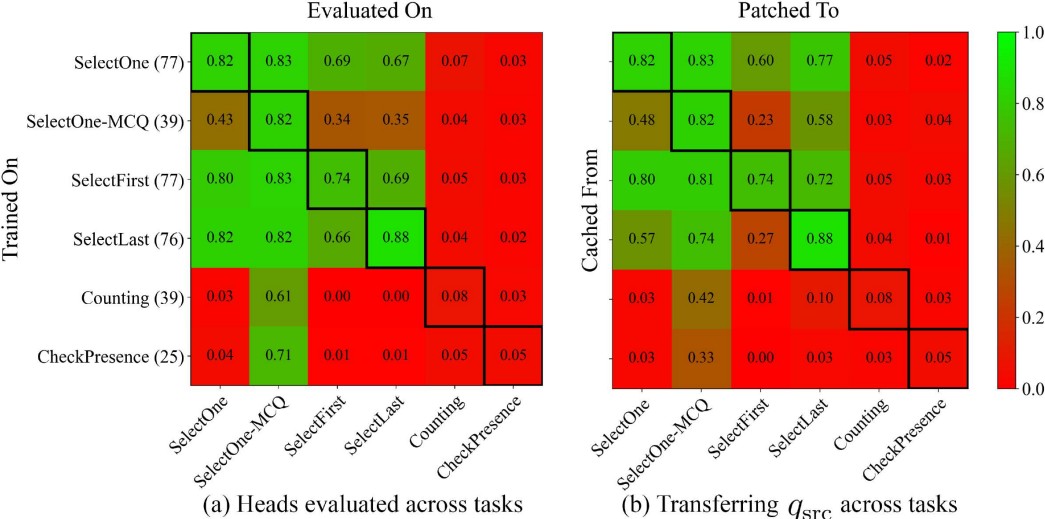

(a) Heads evaluated across tasks     (b) Transferring $q_{\text{src}}$ across tasks

Figure 21: Generalization of filter heads across different tasks. Compare with Figure 3. The figure on the left shows if the same filter heads generalize across tasks. And the figures on the right shows if the predicate can be transferred to a different task.

## L.4 DUAL IMPLEMENTATION IN QUESTION BEFORE VS QUESTION AFTER

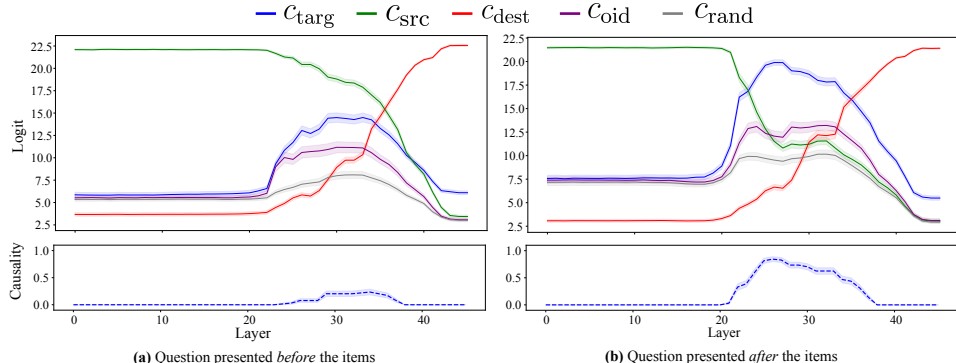

**(a)** Question presented *before* the items

**(b)** Question presented *after* the items

Figure 22: Effect of patching the residual latents in Gemma-27B. Compare with Figure 8.

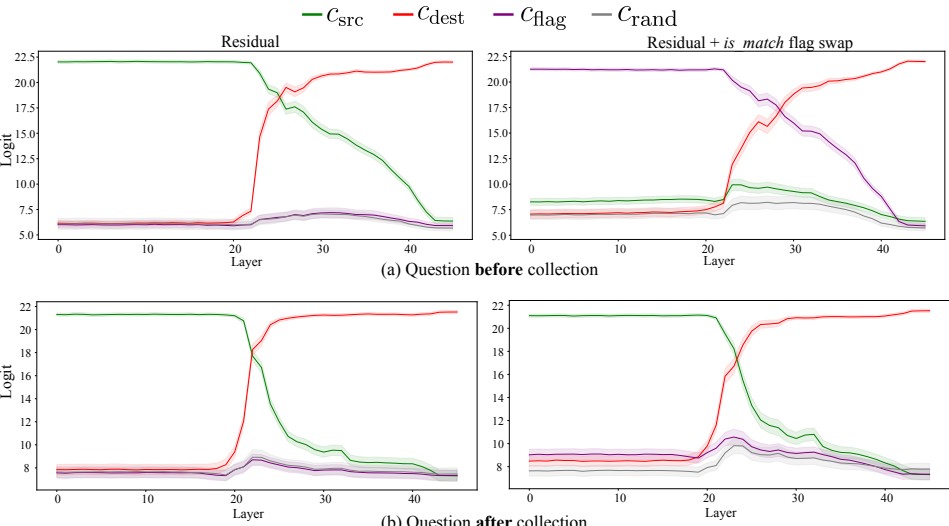

**(a)** Question **before** collection

**(b)** Question **after** collection

Figure 23: Effect of swapping the *is_match* flag between items in Gemma-27B. Compare with Figure 11.

## M  FILTER HEADS IN SMALLER LMS: LLAMA-8B AND GEMMA-9B

We investigate whether filter heads also emerge in smaller models from the same families: Llama-8B and Gemma-9B. We find that filter heads do exist in these smaller models, though they show slightly weaker cross-task generalization compared to their larger counterparts (Llama-70B and Gemma-27B). We hypothesize that this degradation likely because of the increased parameter constraints in smaller models individual attention heads are more inclined to serve multiple computational roles, causing the predicate signal to become more entangled with other information. However, these results hint that the filter head mechanism is architecturally general in Llama and Gemma family of LMs, but its abstraction and portability improve with model scale.

Table 15: Causality of filter heads on *SelectOne* tasks. Using object-type filtering queries we locate 42 heads in Gemma-9B and 47 heads in Llama-8B. We check if the same heads generalize to semantically different predicates like profession identification. Compare with Llama-70B in Table 1 and Gemma-27B in Table 13.

| Semantic Type | Model | |
|---|---|---|
| | Gemma-9B (42 Heads) | Llama-8B (39 Heads) |
| Object | 0.856 (438/512) | 0.826 (423/512) |
| Person Profession | 0.828 (424/512) | 0.756 (387/512) |
| Person Nationality (+ Priming Prefix) | 0.387 (198/512) | 0.525(269/512) |
| Landmark in country (+ Priming Prefix) | 0.492 (252/512) | 0.492 (252/512) |

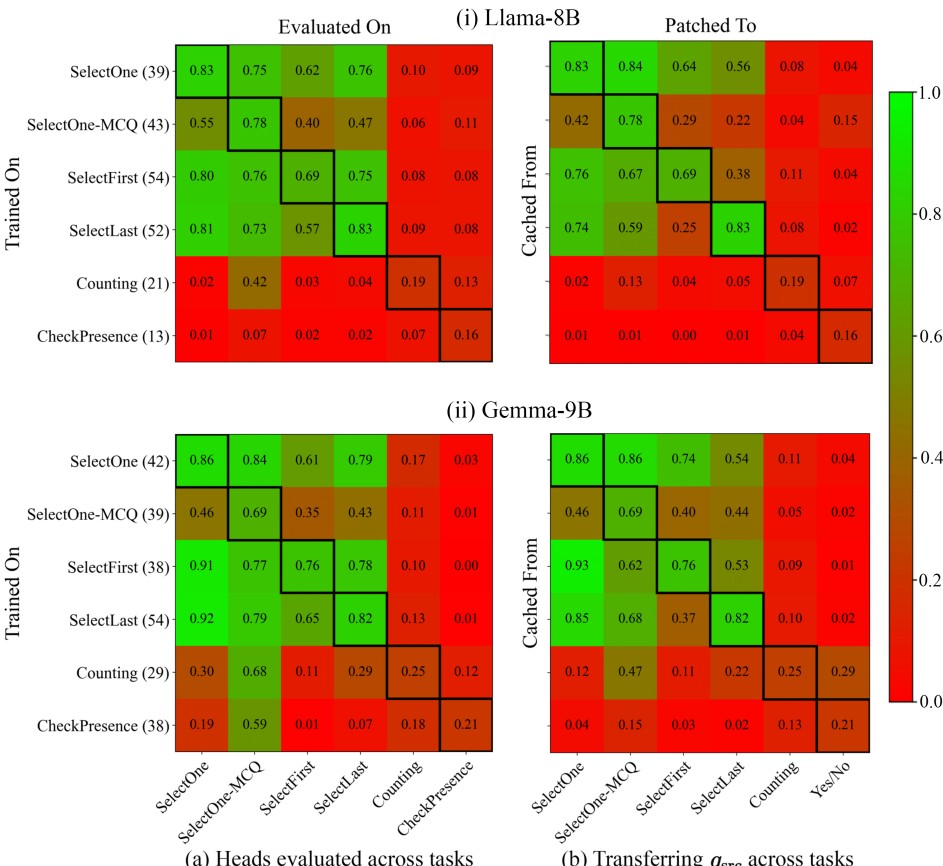

(a) Heads evaluated across tasks       (b) Transferring $q_{src}$ across tasks

Figure 24: Generalization of filter heads across different tasks on Llama-8B and Gemma-9B. Compare with Figure 3 for Llama-70B and Figure 21 for Gemma-27B.

# N    USAGE OF LLMS

Proprietery LLMs services with black-box access such as `claude.ai` and `gemini.google.com` were used as a general purpose assist tool, which is allowed as per the ICLR 2026 author guideline. We have used such LLMs to polish some of the writings in this paper. We have also used LLMs to get more items for our dataset and translating the prompts to other languages.

# O    LIMITATIONS

Our investigation of filtering mechanisms in LLMs, while revealing important insights, has several limitations

**Task Coverage.**    We examined only six filter-reduce tasks, which may not capture the full diversity of filtering strategies employed by LLMs. Even within our six tasks we identified that the filter heads do not show high causality in the *CheckPresence* task, indicating that the LM uses alternate mechanisms for certain filtering operations. The consistent prompt templates we used enabled us to scale up our controlled experiment setup, but they may have biased us towards specific computational strategies inside the LM. LMs may adapt their filtering approach based on what information is available in ways that our limited task set and prompting strategies cannot fully reveal.

**LM Coverage.**    We identified filter heads in Llama-70B, Llama-8B, Gemma-27B, and Gemma-9B models. The fact that we were able to identify similar mechanisms in four models of different sizes, from different families, trained on different datasets echoes the idea of Evolutionary Convergence from Morris (2006): distantly related organisms (e.g. vertebrate and cephalopods) independently evolve similar adaptations (e.g. eyes) in response to similar environmental pressure.

However, such convergence is not guaranteed across all LMs. Notably, findings from Zhong et al. (2023) suggest that identical architectures trained on different datasets can potentially develop different implementations for the same operation. LMs from other families may develop a mechanism that does not make use of such filter heads. Additionally, we restrict our analysis to fairly larger LMs. It is possible that smaller LMs, where parameter constraints might enforce higher head-level superposition, the predicate representation may get entangled with other unrelated stuff in the query states. Therefore, we might not see the distinct head-level causal role of filter heads we get in larger LMs.

**Implementation.**    Our tasks are designed such that we can determine whether an answer is correct based on what the LM predicts as the next token. While curating a prompt we ensure that none of the items share a first token with each other. In all of our experiments we also ensure that the answer of the source prompt, destination prompt, and the target answer of the transported predicate are all different, in order to ensure that patching does not merely copy over the answer. However, this choice of validating with only the first predicted token has potentially restricted us from exploring other tasks that have a similar filter-reduce pattern of operation. Most of the causality scores we report in this paper were calculated on a single trial with 512 examples sampled randomly. It is possible that these scores would change slightly on a different trial.

# P  QUALITATIVE ATTENTION PATTERNS OF FILTER HEADS

In this section we provide the aggregated attention pattern of the filter heads in different examples. We use the same 79 filter heads that we identified in Llama-70B for the SelectOne task. The query token is the last token in all cases.

## P.1  TYPICAL FILTER HEAD BEHAVIOR ON TASKS WE INVESTIGATE

Filter heads exhibit a distinct attention pattern: when given a filtering task, they selectively focus their attention on items that satisfy the filtering criteria. Below we visualize this behavior across the six tasks we study in the main text.

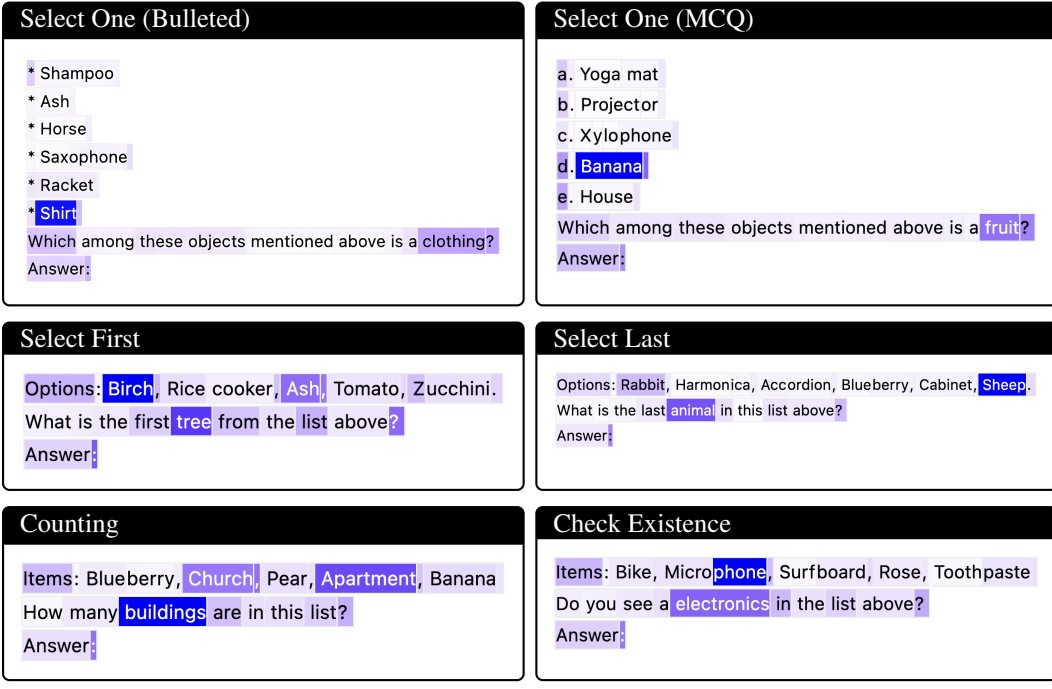

## P.2 GENERALIZATION BEYOND OUR CORE TASKS

To check the generalization of filter heads beyond our six core tasks, we examine their attention patterns on additional filtering scenarios that we did not systematically evaluate in the main text. These examples include negated predicates, logical deduction, mathematical reasoning, and identifying odd-one-out items.

Across these diverse scenarios, we observe that the same 79 heads identified for the *SelectOne* task maintain their characteristic filtering behavior: their aggregated attention consistently focuses on items satisfying the given predicate. This supports our claim that a range of different filtering operations share a common computational circuit implemented by a consistent set of filter heads.

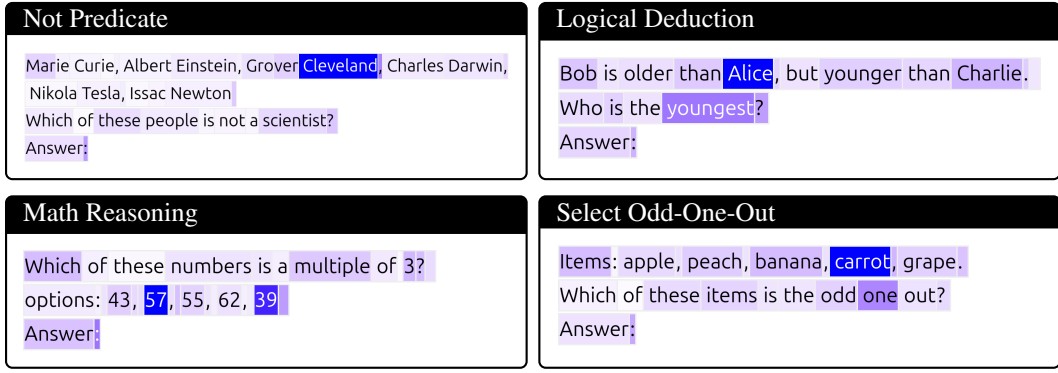

## P.3 FAILURE CASES OF FILTER HEADS

We also identify boundary cases where filter heads do not exhibit their characteristic attention patterns. These tasks require reasoning with non-semantic information—such as phonological similarity, letter counting, alphabetical ordering. This suggests that filtering operations requiring non-semantic reasoning may rely on different computational mechanisms. These alternate mechanisms may not involve the filter heads we identified or these heads may play some other role which we have not investigated within the scope of this work.

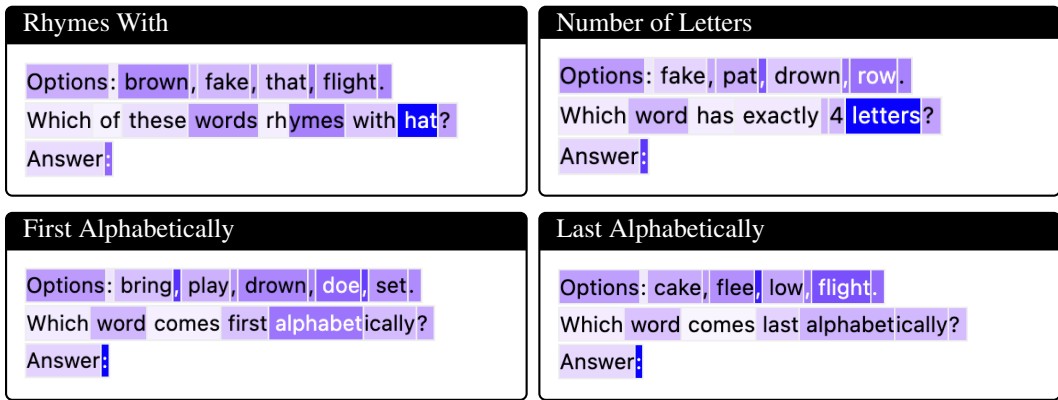

## P.4 APPLICATION

We demonstrate a practical utility of filter heads in detecting the presence of certain concepts in the text: identifying false information and detecting sentiment.

We curate paragraphs/free-form texts that mix factual and false statements about a topic. We break the text into sentences and then append a question asking the LM to identify the false information. When we visualize the aggregated attention pattern of the filter heads, identified with the *SelectOne* task, we observe that the heads focus their attention on the last token of the sentences containing false information. We annotate the last tokens of the sentences with a black border to aid visualization.

We can apply the same approach for sentiment analysis in movie/item reviews containing both positive and negative sentences.

---

**Lie Detector 1**

Carl Sagan was an American astronomer, astrophysicist, and science communicator.
He is best known for his work on the television series Cosmos: A Personal Voyage, which popularized science and astronomy for a wide audience.
Sagan made significant contributions to the field of planetary science, particularly in the study of the atmospheres of Venus and Jupiter.
He was awarded the Pulitzer Prize for his book "The God Delusion" in 1978.
Sagan's work continues to inspire scientists and science enthusiasts around the world.

---

**Lie Detector 2**

ET is a movie about an alien who is stranded on Earth.
He befriends a young boy named Elliott and they form a close bond.
Together, they embark on a journey to help ET return home.
It is considered a classic in the romantic comedy genre.
The film explores themes of friendship, adventure, and the importance of family.
ET's iconic appearance and memorable quotes have made it a beloved film for audiences of all ages.
The movie was directed by Christopher Nolan.
The movie received critical acclaim for its heartwarming story and special effects.

---

**Negative Sentiment Detector 1**

Jaws is a boring movie about a shark terrorizing a town.
The music is iconic and builds suspense well.
The effects still hold up as moving today.
The characters however lack depth and are poorly acted.
The final scene is thrilling and satisfying.
It is a classic that still holds power today.

---

**Negative Sentiment Detector 2**

I bought this keyboard for my IPad a few weeks ago.
The keys are very responsive and have a nice tactile feel.
The battery life is quite disappointing, lasting only about 4 hours on a full charge.
The build quality is solid and the design is great.
Overall, it's a decent keyboard but has some drawbacks that prevent it from being perfect.

---

## Q  FILTER HEADS IN LLAMA-70B

| Layer | Head | Indirect Effect |
|-------|------|-----------------|
| 35 | 19 | 3.546021 |
| 39 | 45 | 1.353394 |
| 35 | 17 | 1.306396 |
| 31 | 38 | 1.114380 |
| 35 | 40 | 0.611328 |
| 35 | 20 | 0.443115 |
| 31 | 39 | 0.340698 |
| 35 | 18 | 0.281738 |
| 29 | 56 | 0.208984 |
| 42 | 31 | 0.154541 |
| 28 | 40 | 0.151733 |
| 29 | 61 | 0.141235 |
| 36 | 47 | 0.128540 |
| 34 | 6 | 0.110474 |
| 37 | 30 | 0.106079 |
| 35 | 23 | 0.104248 |
| 31 | 33 | 0.098511 |
| 33 | 18 | 0.098145 |
| 29 | 57 | 0.076416 |
| 37 | 39 | 0.073608 |
| 34 | 33 | 0.068726 |
| 35 | 27 | 0.052734 |
| 35 | 28 | 0.052734 |
| 28 | 45 | 0.050049 |
| 33 | 30 | 0.042725 |
| 39 | 35 | 0.038818 |
| 38 | 19 | 0.028809 |
| 38 | 49 | 0.022949 |
| 36 | 44 | 0.022461 |
| 36 | 17 | 0.018066 |
| 50 | 34 | 0.017456 |
| 36 | 54 | 0.017090 |
| 37 | 36 | 0.016479 |
| 37 | 16 | 0.011108 |
| 36 | 52 | 0.010376 |
| 36 | 22 | 0.003052 |
| 32 | 12 | 0.001953 |
| 38 | 51 | -0.001221 |
| 45 | 1 | -0.001953 |
| 37 | 7 | -0.003296 |

| Layer | Head | Indirect Effect |
|-------|------|-----------------|
| 35 | 5 | -0.003418 |
| 39 | 36 | -0.003418 |
| 30 | 62 | -0.004272 |
| 32 | 48 | -0.004395 |
| 31 | 40 | -0.005005 |
| 35 | 36 | -0.011719 |
| 32 | 19 | -0.013794 |
| 33 | 23 | -0.014160 |
| 33 | 46 | -0.017456 |
| 37 | 3 | -0.021362 |
| 29 | 62 | -0.030029 |
| 47 | 17 | -0.036133 |
| 31 | 0 | -0.036743 |
| 38 | 50 | -0.042847 |
| 42 | 30 | -0.043091 |
| 31 | 37 | -0.055786 |
| 37 | 0 | -0.061890 |
| 33 | 21 | -0.063599 |
| 37 | 4 | -0.067749 |
| 36 | 40 | -0.068481 |
| 49 | 1 | -0.069824 |
| 35 | 22 | -0.079346 |
| 29 | 60 | -0.085938 |
| 49 | 7 | -0.087402 |
| 33 | 43 | -0.096558 |
| 31 | 36 | -0.100586 |
| 31 | 32 | -0.109375 |
| 49 | 5 | -0.162842 |
| 42 | 28 | -0.172974 |
| 31 | 43 | -0.183960 |
| 37 | 28 | -0.188232 |
| 49 | 4 | -0.216553 |
| 34 | 1 | -0.247803 |
| 38 | 23 | -0.250610 |
| 34 | 45 | -0.252563 |
| 47 | 18 | -0.437988 |
| 39 | 44 | -0.486816 |
| 35 | 42 | -0.770020 |
| 39 | 41 | -0.843811 |

Table 16: Indirect effect scores for filter heads, sorted in descending order.

