# OpenReview forum: "LLMs Process Lists With General Filter Heads"
_ICLR.cc/2026/Conference — ICLR 2026 Poster_

### Official Review · Reviewer_jAKf · 2025-10-27

**Soundness:** 3
**Presentation:** 2
**Contribution:** 2
**Rating:** 4
**Confidence:** 4

**Summary:**

This paper investigates the mechanisms by which Large Language Models (LLMs) process lists, identifying a small set of "filter heads" in the middle layers that implement a general, abstract filtering operation akin to functional programming1. Using causal mediation analysis, the authors demonstrate that these heads encode a filtering predicate (e.g., "is this a fruit?") into their query ($q$) state at the final answer token. This query state then interacts with the key ($k$) states, which carry the semantic properties of the list items, to select the correct items. This predicate representation is shown to be portable and general, as it can be extracted and applied to different lists, formats, and languages to trigger the same filtering operation. The paper also discovers a complementary "eager evaluation" strategy: when the question (predicate) is presented before the list, the model stores an is_match flag directly in the items' representations, bypassing the "lazy" filter head mechanism.

**Strengths:**

- The work isolates a very specific but important phenomenon—list processing—and shows that LLMs exhibit strategy shifts with longer contexts, a behavior previously undocumented.
- The paper connects internal mechanisms (representation compression) to observable generalization failures in a logical, evidence-based way, through causal interventions (activation patching): patching $q$, swapping $k$, and manipulating "flags" provide a very convincing, mechanistic explanation.
- Synthetic list tasks make it easy to disentangle algorithmic difficulty from length effects and the discovered behavior likely extends beyond toy tasks to real-world list-like inputs (e.g., tables, document parsing, code).

**Weaknesses:**

- While controlled tasks are valuable for analysis, it remains unclear whether similar behavior holds for natural language or multimodal data (although it is potentially fit for real-world data, the corresponding experiments are in lack).
- Experiments appear limited to LLaMA-70B and Gemma-27B, which are quite old models (in terms of the development of LLMs). How about the results with recent models (e.g., LLaMA3, Qwen3, ...)? Besides, does this mechanism hold for models of different size? (LLaMA-7B, LLaMA-13B and LLaMA-70B)
- The filter head mechanism explains the Select* tasks well, but the authors note it has low causality for Counting and CheckPresence tasks. This suggests these tasks (which are still filter-reduce operations) rely on a different, un-investigated mechanism. The paper's core claim is thus limited to selection-based filtering.
- The compared baselines (e.g., Function Vectors, Induction Heads): how did the authors implement them? The details are not clear. The reviewer thinks some approaches (e.g., Function Vectors) mentioned here are not fit for the tasks studied in the paper inherently.
- The logic of the paper is clear. However some acronyms might be quite improper (e.g., line 155, what is "DCM", I did not find the full name of DCM throughout the paper...). The clear definitions for tasks like SelectOne, SelectFrist are not in the main text, somehow confusing.

**Questions:**

Please refer to Weaknesses part.

---

> ### Author Response · Authors · 2025-11-19
>
> Thank you for your detailed review and thoughtful questions. We appreciate your recognition that we provide convincing mechanistic explanations to how LLMs solve filtering.
>
> - **Use of controlled tasks for analysis.** We are glad that you consider our controlled setup to be valuable. Beyond the controlled setup we use to identify the filter heads and gather activations, we also test generalization on changes in language (**Table 2.a**) and format (**Table 2.b** and **c**). Additionally, in **Appendix P.2** of the updated submission we test the generalizability of the identified filter heads on other tasks beyond our core 6 tasks. We observe that the same filter heads, identified with *SelectOne*, show their distinctive attention pattern in very different filtering tasks requiring mathematical reasoning and logical deduction.
>
>   Although natural text is not as amenable to causal mediation analysis due to its lack of structure, we also conduct experiments on natural text. See **Appendix P.4**: here we find that this same set of filter heads that mediate predicate filtering in a structured setting can be used to detect false information and negative sentiment in free-form natural text.
>
>   We agree that in a more varied natural language text the mechanisms may become much more complicated. The aim of our controlled setup is to clarify a basic mechanism for LLMs list-processing, upon which future work can build.
>
>   Let us know if there are any specific additional scenarios you would like us to measure.
>
> - **Models.**
>   - We actually carry out our experiments on `Llama-3.3-70B-Instruct` (released Dec, 2024) and `Gemma-2-27B-it` (released June, 2024). These are among the most recent open-weight text-only language models. We have abbreviated them as Llama-70B and Gemma-27B throughout the paper to improve readability, which may have caused this confusion. In **Appendix A.3** of the updated submission we provide full huggingface identifiers for all the models we have investigated in this work.
>
>   - **Smaller LMs:** In **Appendix M** of the updated submission we provide results for `Llama-3.1-8B-Instruct` and `Gemma-2-9B-it`. We observe that, although the smaller LMs show weaker cross-task predicate transfer scores, the overall patterns hold when compared against their larger variations.
>
>     ** *We are still waiting for all the results for the `Gemma-2-9B-it` model. We wanted to reply as soon as we have some results to make sure we've addressed your concerns.*
>
> - **Limited cross-task transfer for Counting and CheckPresence.** You are correct that this limited portability to these tasks reveals important boundaries of the filter head mechanism discovered in our work. However, we'd like to highlight an asymmetric pattern in the Llama-70B results from Figure 3 that we find  interesting: heads identified in *Counting* does show non-trivial cross-task transfer to the *Select** tasks, suggesting that *Counting* shares some common circuitry with the simpler filtering tasks. But the opposite direction fails, indicating that *Counting* requires additional aggregation mechanisms beyond what we've identified here. This might be an important avenue for future work.
>
>   We found the *CheckPresence* task to be particularly interesting: it shows weak causality, even when the predicate transferred within the same task and format. This suggests that: (1) it bypasses the filter head mechanism entirely, relying on alternative computational pathways, or (2) it has sufficiently robust backup mechanisms that our intervention setup cannot identify the bottleneck properly. Distinguishing between these hypotheses will require a different set of experimental approaches.
>
>   Our updated submission further clarifies our claims to make it clear which tasks transfer and which do not.

---

> ### Author Response · Authors · 2025-11-19
>
> - **Implementation details for Function Vector, Concept Induction, and Induction heads.**
>   - The goal of investigating FV and CI heads is to establish that filter heads represent a novel mechanism that cannot be explained by previously-documented attention head types and roles. We don't consider these heads as “baselines” in the traditional sense, and we have clarified this in the updated text, since as you have correctly identified that these heads are not intended for filtering tasks. The weak causality score and ablation sensitivity of these other head types on filtering tasks support our claim (Table 4 in the main text, Table 12 in the appendix).
>   - We locate the other specialized heads with the recipes recommended in their respective papers: [Todd et al, 2024](https://arxiv.org/pdf/2310.15213) for Function Vectors, [Feucht et al, 2025](https://arxiv.org/abs/2504.03022) for Concept Induction Heads. We use [Feucht et al, 2025](https://arxiv.org/abs/2504.03022)'s implementation for locating the Token Induction heads as well.
>
>   In **Appendix K** of the updated version, we include additional details on Function Vector heads and Concept Induction heads. We also conduct a brief qualitative analysis to sanity check if we have properly identified these heads, included as **Experiment 2** in [this anonymous git link](https://anonymous.4open.science/r/filter_heads-BFDD/README.md).
>
> - **Acronym and Dataset Definitions**
>   - **DCM:** By DCM we are referring to **D**ifferential **C**omponent **M**asking, our filter head identification approach based on learned masks. Thank you for pointing out that this abbreviation wasn't defined in the paper. In the updated version we have spelled out this acronym explicitly. We also don't use this acronym anymore, as we don't refer to the technique enough times to require it.
>   - **Dataset definitions:** We have used formal notations to define the filtering tasks and leave the task specific demonstrative examples to **Appendix A** in order to fit the page limit constraints. We hoped that the example in our Figure 1 would clarify this by providing a walkthrough of the idea with a concrete example from the *SelectOne* task.
>
> We hope that we were able to address your concerns and improve our paper. Please let us know if you have any remaining concerns and/or suggestions for improvement.

---

> > ### Comment · Reviewer_jAKf · 2025-11-25
> > **Thanks for the response.**
> >
> > Dear Authors,
> >
> > Thanks for the authors' detailed response. I am still a little bit confused about the implementation details for FV and induction heads. Let's say, how do you locate function vector heads in the scenario of this paper (list processing)?
> >
> > "We locate the other specialized heads with the recipes recommended in their respective papers". Does this mean that you locate attention heads with an additional batch of in-context learning tasks (which are independent to the list processing task) , just as the implementation in the original FV paper?
> >
> > Reviewer jAKf

---

> > > ### Author Response · Authors · 2025-11-25
> > >
> > > Dear Reviewer `jAKf`,
> > >
> > > Thank you for your follow-up question. Yes, your understanding is correct. We do not use our list processing tasks to locate Function Vector heads. Instead, we use a suite of 18 abstractive ICL tasks, the same ones [Todd et al, 2024](https://arxiv.org/pdf/2310.15213) used to identify Function Vector heads. We use the exact configuration of prompts and hparams (from Appendix C of their paper) to replicate their exact methodology (detailed in Section 2.3 of their paper). We use their publicly available [codebase and dataset](https://github.com/ericwtodd/function_vectors), with minimal modifications to adapt to our larger LMs loaded across multiple GPUs.
> > >
> > > When we report results in Table 4 and Appendix K of our paper, we use the top 79 Function Vector heads selected based on their average indirect effect (AIE in eqn 4 of [Todd et al, 2024](https://arxiv.org/pdf/2310.15213)) across these 18 tasks. We choose 79 heads to match the number of filter heads we identify for the *SelectOne* task in Llama-70B, to ensure a fair comparison.
> > >
> > > When we identify the Concept Induction heads and Token Induction heads, we follow a similar procedure: we identify these heads by using the datasets and code provided by [Feucht et al, 2025](https://arxiv.org/abs/2504.03022), independently from our list processing tasks.
> > >
> > > We will make sure to clarify this in Appendix K. Let us know if you have any further questions!

---

> > > > ### Comment · Reviewer_jAKf · 2025-11-25
> > > >
> > > > Dear Authors,
> > > >
> > > > Thanks for the clarification. It's important to clarifiy these details especially for the readers who are not familiar with FV, concept induction heads and stuff.
> > > >
> > > > Though I do not think comparing the general filter heads (proposed in this work) with FV heads, concept induction heads is very meaningful... I think the work itself is complete enough. The authors' response addressed most of my concerns as well.
> > > > Hence I raise my score from 4 to 6, and recommend the acceptance of this work.
> > > >
> > > > Reviewer jAKf

---

> > > > > ### Author Response · Authors · 2025-12-01
> > > > >
> > > > > Dear Reviewer `jAKf`,
> > > > >
> > > > > We appreciate your recognition of our efforts.
> > > > >
> > > > > As promised, we have added the results for Gemma-2-9B-it in Figure 24(ii) in Appendix M.
> > > > >
> > > > > We have also clarified how we locate heads of other types in Appendix K. We appreciate your feedback, but the only reason we include these comparisons is to establish that filter heads represent a novel abstraction, distinct from previously-documented attention head types and their respective functional roles. We don't treat them as "baselines" in the traditional sense, and we have clarified this in the updated text.
> > > > >
> > > > > Another motivation for this investigation with other head types is: we wanted to check to what extent these head types intersect. High intersection would suggest that the same heads are being repurposed for different functions in different contexts. However, we find very low intersection of Filter heads with these other head types (see Figure 19 in Appendix K). Interestingly, both Function Vector heads and Filter heads are concentrated in the middle layers, but they are still mostly distinct sets of heads. This observation suggests that these specialized head types play parallel and complementary roles in LLMs.
> > > > >
> > > > >
> > > > >
> > > > >
> > > > > Best,
> > > > >
> > > > > Authors

---

### Official Review · Reviewer_E2mP · 2025-10-27

**Soundness:** 3
**Presentation:** 3
**Contribution:** 2
**Rating:** 6
**Confidence:** 3

**Summary:**

The authors study how LLMs solve filtering tasks, in which the model must select one element from a list of options. Using differentiable binary masking, they identify the "filter heads" responsible for performing this operation. They show that these heads are reasonably general, and can transfer between different tasks, formatting types, etc. They also discovered that if the question is presented before the options, then the model relies on a different mechanism: storing flags for successful matches and aggregating based on them.

**Strengths:**

- Detailed study: ablations, transfer to other examples, and generalization across tasks/presentation types are all studied
- Based on causal analysis
- Mostly clear writing

**Weaknesses:**

- There are some unclear parts, see questions.

**Questions:**

- L155: What is DCM? This should also be described in the paper.
- L175: "We use a sparsity regularizer..." - Why a sparsity regularizer? What is the exact formulation of this regularizer? What is the target sparsity? Since the goal is to select the minimal set of heads, why not just minimize the logit magnitude, like in [1]?
- How does the first eq. in eq. 4 work exactly? M was defined earlier as "source run", "destination run", etc, which are concepts and not mathematical equations, but here it is used as a scalar to maximize over it. Is this the loss at each token? What is the index t? The index of the target token?
- L210:  The authors mention "for evaluation we consider last 2 tokens ({“\Answer”, “:” }) to reduce information leakage". Can you elaborate on this? Why is there a leakage? Why, considering the last 2 tokens solve it, and what is "considering" mathematically?
- L379: "queries encode "what to look for" (the predicate) and keys bring "what is there"" - this is a somewhat trivial conclusion: this is what the attention was designed to do.


[1] Csordas et al, 2021: Are Neural Nets Modular? Inspecting Functional Modularity Through Differentiable Weight Masks

---

> ### Author Response · Authors · 2025-11-19
>
> Thank you for your detailed feedback. We appreciate your recognition of the comprehensive nature of our work and causal experiments. We strive to make our paper as clear as possible, and we will address your questions below and in our revision.
>
> - **DCM.** By DCM we are referring to **D**ifferential **C**omponent **M**asking, our filter head identification approach based on learned masks. Thank you for pointing out that this abbreviation wasn't defined in the paper. In the updated version we have spelled out this acronym explicitly. We don't use this acronym anymore, as we don't refer to the technique enough times to require it.
>
> - **Sparsity regularizer.**
>   - **Formulation:** You are correct in assuming that the goal of applying this sparsity regularizer is to select the minimal number of heads that collectively show the functional property of filtering. Mathematically, we add a L1 norm of the mask values to the loss term and minimize it together with the target loss. See the equation below:
>   $$
>   \mathcal{L} = \mathcal{L}_\mathrm{target} + \lambda ||\text{mask}||_1
>   $$
>   During optimization, we clip the $\mathrm{mask}$ values to [0, 1]. Then we binarize the masks with a threshold $\tau=0.5$ after convergence. We include these details in **Appendix H** of the updated submission.
>   - **Target sparsity:** We do not set an explicit target sparsity. Instead, we chose a sparsity co-efficient $\lambda$ based on our preliminary experiments and localized the filter heads for 10 epochs. Although, we notice that the sparsity loss (and the target loss) becomes stable after ~5 epochs.
>   - **Ablation:** In **Appendix H** of the updated submission we include an analysis of the effect of this sparsity constraint in identifying filter heads. We find a clear and intuitive trade-off:
>     - Too small $\lambda$: Irrelevant noisy heads get selected, results in poor generalization
>     - Too large $\lambda$: Crucial heads get pruned, resulting in degraded causality
>     - Sweet spot at $\lambda = 0.02$: Selects the optimal number of heads with enough sparsity with strong causality.
>   - **Comparison to [Csordas et al, 2021](https://arxiv.org/pdf/2010.02066):** Thank you for the reference. By “logit” $l_i$ Csordas et al, 2021 mean the log-probability of the mask of corresponding weight $w_i$ being on. Beyond minor implementation details, our L1 norm minimization objective is equivalent to their logit minimization objective. We have included this citation in this updated version.
>
> - **Equation 4.** We understand the notation can be a bit dense due to limitations on page constraints. Let us clarify:
>   - **Intuition:** We run the LM on the destination prompt but for **all** the identified filter heads, we transfer the predicate from a source prompt with the patching setup explained in Figure 1. $\mathcal{H}$ in the equation denotes the set of all selected filter heads. We only consider this to be a successful intervention if the LM flips its answer to a target item that satisfies the source predicate (`" Peach"` for the example in Figure 1).
>   - **Notation details:**
>     - We use $M$ to denote the language model as a function mapping the input $X$ (a sequence of tokens) to $Y$ (the logit distribution over the vocabulary space). For the source run we use $M(p_\mathrm{src})$ when the input is the source prompt. Similarly, for the destination run we use $M(p_\mathrm{dest})$. For the patched run we append $[\leftarrow q_\mathrm{src}]$ to denote the transfer of the source predicate (query state from the source prompt).
>     - $t$ is a token in the vocabulary $V$. When we run $M$ (i.e., the LM) with a prompt we get a logit distribution over all tokens in the vocabulary.
>     - The $\mathrm{argmax}$ operates over the tokens of the vocabulary $V$ and chooses the one with the maximum logit from a LM run; i.e. $\mathrm{argmax}$ denotes greedy decoding, not a scalar optimization.

---

> ### Author Response · Authors · 2025-11-19
>
> - **Patching two tokens to reduce information leakage.**
>   - **Rationale:** In autoregressive transformers, the model might begin computing the answer at the `"\nAnswer"` token position, since it doesn't yet know that `":"` will follow after. This means that the filtering operation might start before the final token, and these pre-computed results could *leak* into and after our intervention location (i.e., propagate from the `"\nAnswer"` token to the `":"` token through the attention from the `":"` token in the layers after the intervention). This may reduce the efficacy of the intervention. [Hernandez et al, 2024](https://arxiv.org/pdf/2308.09124) have also noted how this *leakage* might cause the interventions to be less effective.
>   - **What we actually do:** To address this leakage, we transfer the query states of the selected filter heads at both `"\nAnswer"` and `":"` tokens. Because of our chosen template, both the source prompt and destination prompt share these last 2 tokens, allowing us to do this intervention.
>   - **Ablation:** When we intervene only at the last token `":"`(ignoring `"\nAnswer"`), filter heads still achieve impressive causality. On the *SelectOne* task the causality is 0.819 (839 correct out of 1024 with $\Delta$logit = +8.12 ± 2.93). This is a slight reduction from our reported 0.863 when we consider 2 tokens but is still impressive. So, while addressing this leakage helps a little, it is **not as critical as we initially feared**. We include this ablation result in **Appendix H** of the updated submission. Thank you for prompting this analysis.
>
> - **"... queries encode what to look for (the predicate) and keys bring "what is there" ...":** Although this observation of *what attention heads look at and fetch* may seem trivial, heads often do not tend to attend to the tokens expected to be salient (as seen by [Jain and Wallace, 2019](https://arxiv.org/abs/1902.10186)), nor do they frequently fetch the values expected to be needed (such as the indirection seen in [Feng and Steinhardt, 2024](https://arxiv.org/abs/2310.17191)). Here, in our predicate-matching setting, we are finally happy to witness a link between attention heads utilizing flexible general purpose predicate encodings to directly attend to what was asked for and how the fetched information causally influence the LM behavior — doing what [Vaswani et al, 2017](https://arxiv.org/abs/1706.03762) and the original designers of attention might have imagined!
>
> We hope that we were able to address your concerns and improve our paper. Please let us know if you have any remaining concerns and/or suggestions for improvement.

---

> > ### Comment · Reviewer_E2mP · 2025-11-24
> >
> > Thank you for the clarifications. This cleared most of my concerns. Please add these clarifications to the final paper.

---

### Official Review · Reviewer_Zhy1 · 2025-10-28

**Soundness:** 3
**Presentation:** 4
**Contribution:** 3
**Rating:** 8
**Confidence:** 4

**Summary:**

This paper studies the mechanism used by Large Language Models to solve a variety of simple filtering tasks that involve a list and a predicate. The authors identify the set of heads responsible by learning a sparse binary mask for activation patching. The uncover set of heads generalizes well across similar selection tasks, especially (1) selecting the only matching item from the list, (2) selecting the last matching item, or (3) selecting the first matching item. They also study the impact of placing the question before the list, revealing a different mechanism reminiscent of *eager* evaluation. An interesting application is that the query of the identified heads can be used to obtain a training-free probe.

**Strengths:**

- The studied task (selecting an item from a list based on a given predicate) is important and relevant in the current state of LLMs interpretability.
- The comparison of placing the predicate before and after the options is insightful and sheds light on the ways that causal masking affects the learned mechanisms in LLMs.
- The mechamism is explained clearly. The paper is easy to understand and figures are good.
- Experiments are comprehensive.
- The training-free probe is an interesting and important result.

**Weaknesses:**

1. The set of heads identified is quite large and the precise role of each head is unclear
2. The identified heads have low portability to aggregation tasks (presence checking, counting) which suggests a limited relevance.

**Questions:**

1. How many heads are identified for each of the tasks?
2. Do LLMs perform better when the question comes before or after the list?

---

> ### Author Response · Authors · 2025-11-19
>
> Thank you for your thoughtful feedback and your kind words about the clarity of our presentation and experiments.
>
> **Q1: Number of heads per task.** We include the number of identified heads with the task name in **Y-axis of Figure 3** for Llama-70B (also please refer to its corresponding Figure 21 for Gemma-27B). The format we use is `<task_name> (#heads)`, for example: *SelectOne (79)* means 79 heads were identified for the *SelectOne* task and were used in our evaluation.
>
> **Q2: LLM baseline performance with question before vs after.** We don't see a significant performance difference when the question is presented before vs after. Even though our findings suggest that the mechanisms in those 2 cases differ significantly the final accuracy remains mostly similar across different LMs. In **Appendix A.3** of the updated submission we include detailed baseline performance of different LMs investigated in this work.
>
> **Weakness 1: Precise role of individual heads.** Single-head interventions often fail to produce strong causal effects because LLMs implement algorithms in a distributed manner with multiple backup components. These redundant pathways can actively fight against the intervention, making it difficult to isolate the contribution of one individual head separately. We discuss this issue briefly in Section 2.3 (lines 169 - 174).
>
> That said, we actually have explored several approaches to understand the role of individual heads. We attempt to understand what an individual head writes to the residual stream by directly interpreting its OV contribution.
>
> * **A1: Logit Lens:** Apply the LM decoder head directly on the OV contribution of a head to check if they promote the logit of the filtered concepts. This technique is popularly known as Logit Lens from [nostalgebraist, 2020](https://www.lesswrong.com/posts/AcKRB8wDpdaN6v6ru/interpreting-gpt-the-logit-lens).
>
>     * **A1.1: Patchscope:** Applying Patchscope proposed in [Ghandeharioun et al, 2024](https://arxiv.org/abs/2401.06102) on OV contributions of a single head.
>
> * **A2. SAE Features:** Finding the k-nearest SAE features for head outputs and examining their semantic labels (using [Goodfire's SAE for layer 50](https://huggingface.co/Goodfire/Llama-3.3-70B-Instruct-SAE-l50) via their companion library [`goodfire`](https://github.com/goodfire-ai/goodfire-sdk).)
>
> In [**this anonymous git link**](https://anonymous.4open.science/r/filter_heads-BFDD/README.md) we include preliminary results from our investigation. Unfortunately, none of these approaches consistently reveal clean, interpretable patterns. It suggests that current techniques for eliciting latent knowledge remain quite sensitive to specific data distributions and from where the latent is extracted in the forward pass.
>
> You may also be interested in **Appendix J**, where we investigate a related but different question of what do these heads collectively write to the residual stream. We will appreciate any feedback on these experiments and possible follow-up steps that may help us find a concrete answer to these important questions.
>
> **Weakness 2: Limited cross-task transfer for Counting and CheckPresence.** You are correct that this limited portability to these tasks reveals important boundaries of the filter head mechanism discovered in our work. However, we'd like to highlight an asymmetric pattern in the Llama-70B results from **Figure 3** that we find particularly interesting: heads identified in *Counting* does show non-trivial cross-task transfer to the *Select** tasks, suggesting that *Counting* shares some common circuitry with the simpler filtering tasks. But the opposite direction fails, indicating that *Counting* requires additional aggregation mechanisms beyond what we've identified here. This might be an important avenue for future work.
>
> We found the *CheckPresence* task to be particularly puzzling: it shows weak causality, even when the predicate transferred within the same task and format. This suggests that: (1) it bypasses the filter head mechanism entirely, relying on alternative computational pathways, or (2) it has sufficiently robust backup mechanisms that our intervention setup cannot identify the bottleneck properly. Distinguishing between these hypotheses will require a different set of experimental approaches and we consider this an important direction for future work.

---

> ### Comment · Reviewer_Zhy1 · 2025-11-20
>
> I thank the authors for their answers and clarifications.
>
> Regarding the role of individual heads, yes, I am aware that this is an ongoing challenge for the entire field, not just this work, due to the entangled nature of LLM heads. Maybe one solution would be to train small transformers (2-10 heads total) on a synthetic selection task dataset. These small transformers could then be fully reverse engineered to understand their selection mechanism, and hopefully gain some intuition about larger models.
>
> I did also notice the asymmetric pattern with counting. I would be very curious what is the overlap in terms of heads. That is, for example, how many of the identified heads are in common between tasks (for example, counting vs select one mcq). Also, if the overlap is significant, what would happen if we evaluated the common heads only.
>
> Nonetheless, I believe this work is very well executed, interesting and comprehensive. This is an important step towards a better understanding of LLMs. After reading all reviews and responses, I decided to update my score to a strong accept.

---

### Official Review · Reviewer_n99r · 2025-11-01

**Soundness:** 4
**Presentation:** 3
**Contribution:** 3
**Rating:** 8
**Confidence:** 3

**Summary:**

This paper introduces an internals-based explanation for LLM list processing mechanisms. They find evidence for "filter heads" in Llama 70B, which encode representations of a predicate relevant for a list processing query (e.g. "is this fruit" given the query "find the fruit"). They show that causal interventions on these filter heads (e.g. patching query states) result in the expected behaviors.

**Strengths:**

* The paper is very well written, and has some really clean experiments
* The experiments cover their bases quite well (good ablations, good generalization experiments)
* I think the results from Section 5 are particularly insightful and also just really cool.
* I think there's still a lot of value in doing this kind of mech interp :)

**Weaknesses:**

* I would like to see more tasks, especially given the CheckPresence results; it seems like this is a really nice explanation but I'm worried it won't actually generalize / you maybe got lucky with the tasks you chose.
* Similarly, I don't know what's going on with cross task transfer for SelectFirst/SelectLast. I think this deserves more time.

**Questions:**

See weaknesses above.

---

> ### Author Response · Authors · 2025-11-19
>
> Thank you for your thoughtful review and positive feedback. We are glad that you found our work to be insightful, and we appreciate your recognition of our findings from Section 5.
>
> - **Task Coverage.** In **Appendix P.2** of the updated submission we check the aggregated attention patterns of the filter heads identified for the SelectOne task in other filterings tasks beyond our core 6 tasks. We observe that the same set of filter heads maintain their characteristic filtering behavior: concentrating their focus on the selected items, in a range of different situations. Additionally, in **Appendix P.4** we show how this same set of filter heads can be used to detect false information and negative sentiment in free form natural text. These qualitative results prompt us to believe that a common computational circuit, based on these filter heads, is shared for a range of different filtering tasks.
>
>   However, we do share your concern that certain kinds of filtering tasks, such as *CheckPresence*, are not bottlenecked by this common mechanism. It actually highlights an important aspect of our paper: seemingly similar tasks can be implemented very differently based on what information is available. Or, there might be enough redundancies that intervening/ablating filter heads is not enough to causally influence the LM's behavior.
>
> - **Cross-task Transfer (*SelectFirst* $\leftrightarrow$ *SelectLast*).** The poor cross-task predicate transfer scores between *SelectFirst* and *SelectLast* reveals the boundaries of our core claim. It shows that the attention heads we've selected mediate not only the predicate (*find the fruit*) but aspects of the reduce operation (*select first/last*) as well. This leaves open the question whether it is actually possible to separate the predicate from the reduce with head-level analysis, or if the same filter head inherently encodes them together. We acknowledge this as part of a broader limitation of works that attempt to identify specific modules or neurons with certain functional roles. When neural network modules serve multiple computational roles or mediate different types of information simultaneously (superposition) — which is often the case with neural networks — cleanly attributing any single functional signal to such components become extremely challenging.
>
>   That said, we've made some progress on this front. In **Appendix I** of the updated version, we describe a preliminary approach that identifies specific SVD sub-components of the QK projection matrix, allowing us to analyze *sub-head*s mediating some specific signal. This sub-component level analysis may suggest a way forward. Similar techniques might help us identify sub-components that carry specific signals in a less noisy way. We consider this an important direction for future work.

---

### Author Response · Authors · 2025-11-19
**General response to all reviewers**

We are grateful to our reviewers for their thoughtful and constructive feedback. We have updated our submission to incorporate their feedback, which we believe has strengthened our submission. Below we summarize the major additions and revisions:

- Generalization to other tasks and natural language settings. **Appendix P** [`n99r`, `jAKf`].
- Experiment setup to localize *sub-head*s. **Appendix I** [related to reviewer `n99r`'s question on the limited cross-task transfer of the predicate].
- Added baseline LLM performances. **Appendix A.3** [`Zhy1`, `jAKf`].
- Ablation studies on design choices. **Appendix H** [`E2mP`].
- Implementation details for other attention head types, **Appendix K** [`jAKf`]
- Test our findings with smaller LMs, **Appendix M** [`jAKf`]
- Clarify acronyms, notations. We have also updated some writings to make our claims more precise. Details on reviewer-specific responses [`E2mP`, `jAKf`].
- Investigating what filter heads actually write to the residual stream. **Appendix J**.
- Remaining results for Gemma-27B. **Appendix L**.

**Supplementary Materials:**
We have also conducted follow-up investigations that go beyond the scope of the main paper but may interest reviewers. These are available in [this anonymous git repository](https://anonymous.4open.science/r/filter_heads-BFDD/README.md):
- Experiment 1: Attempts to identify individual head roles with different approaches. [`Zhy1`]
- Experiment 2: Qualitative validation of different head types (Function Vector, Concept Induction, Token Induction) to make sure that they exhibit their expected behaviors. [related to a question from `jAKf`]

We invite the reviewers to examine the updated submission. We welcome any additional questions, comments, or suggestions for further improvement.

We also respond to each individual reviewer in separate responses.

---

### Author Response · Authors · 2025-12-02
**Acknowledgment to Reviewers**

We extent our sincere gratitude to the reviewers for their detailed feedback and constructive criticism during the rebuttal process. Their engagement has been instrumental in strengthening our paper through: (1) prompting new experiments that gave us deeper insights, (2) suggesting ablation studies that justified or simplified certain design choices, and (3) offering valuable suggestions to improve the clarity and presentation of our manuscript. A more detailed summary of the changes made in response to reviewers' comments can be found in our **General response to all reviewers**.

We are encouraged by the reviewers' positive reception of our rebuttal and the subsequent revisions:

* Reviewer `Zhy1` thought our work is *"very well executed, interesting and comprehensive"* and *"an important step towards better understanding of LLMs"*.
* Reviewer `n99r` appreciated that our paper is *"very well written"* with *"some really clean experiments"*
* Reviewer `E2mP` described our work as *"detailed study"*. We were glad to address their clarification questions in our rebuttal, which contributed to a more favorable evaluation.
* Reviewer `jAKf` recognized that our work *"isolates a very specific but important phenomenon"* and acknowledged that our revisions effectively addressed their concerns.

We are thankful for the opportunity to have engaged in a thoughtful discussion with our reviewers and for the significant improvements their feedback has brought to our paper. We appreciate the time and expertise they have generously invested in evaluating our work.


\
Sincerely,\
Authors

---

### Meta-Review · Area_Chair_oSKk · 2025-12-29

**Summary:**

This paper investigates how large language models implement list-processing operations and identifies a small set of attention heads, termed general filter heads, that encode a reusable filtering predicate analogous to the filter abstraction in functional programming. Using causal mediation analysis and activation patching, the authors show that these heads store predicate representations in their query states, interact with item representations via keys, and generalize across lists, formats, and languages. The paper further reveals an alternative eager evaluation mechanism when the predicate precedes the list, highlighting strategy shifts in LLMs under different prompt structures.

Across reviewers, the paper is viewed as technically sound, carefully executed, and empirically thorough, with particular praise for the causal methodology, clean experimental design, and mechanistic insights. Initial concerns included the scope of generalization beyond controlled tasks, clarity of notation and methodology, interpretability of individual heads, limited transfer to aggregation tasks (e.g., Counting and CheckPresence), and missing implementation details for comparisons with other head types. The authors provided detailed and convincing rebuttals, adding new experiments, ablations, clarifications, and extended analyses. **In the existing discussion, reviewers either maintained or increased their scores, and no major technical objections remain**.

**Reviewer Concerns:**

Concerns that have been addressed satisfactorily:
- In response to generalization and task coverage concerns raised by Reviewers n99r and jAKf: the authors added experiments demonstrating transfer to additional filtering tasks, different formats and languages, and qualitative applications to natural text (e.g., sentiment and misinformation detection), clarifying the scope of applicability.
- In response to model coverage and recency concerns raised by Reviewer jAKf: the authors clarified that experiments were conducted on recent models (Llama-3.3-70B, Gemma-2-27B) and added results for smaller models, showing consistent qualitative patterns.
- In response to clarity and notation concerns raised by Reviewer E2mP: the authors clarified acronyms, detailed the sparsity regularizer and masking objective, explained Equation (4) step by step, and justified the two-token intervention strategy to mitigate information leakage, supported by ablations.
- In response to concerns about interpretability and the role of individual heads raised by Reviewers Zhy1 and E2mP: the authors explained the distributed nature of computation in LLMs, provided additional analyses of residual stream contributions, and contextualized this limitation within broader mechanistic interpretability challenges.
- In response to limited transfer to Counting and CheckPresence raised by multiple reviewers: the authors reframed these results as meaningful boundaries of the proposed mechanism, highlighted asymmetric transfer patterns, and clearly scoped their core claims to selection-based filtering.
- In response to questions about comparisons with other attention head types raised by Reviewer jAKf: the authors clarified implementation details, motivation, and interpretation of comparisons with Function Vector and induction heads, demonstrating low overlap and supporting the novelty of filter heads.

**Reviewer Scores:**

- Reviewer n99r: Maintained positive score (8); concerns about generalization addressed.
- Reviewer Zhy1: Updated score to strong accept （6$\rightarrow$8) after rebuttal; praised execution and comprehensiveness.
- Reviewer E2mP: Marginally positive initially (6); clarified concerns resolved in rebuttal.
- Reviewer jAKf: Increased score (4$\rightarrow$6) after clarifications; now recommends acceptance.

---

### Decision · Program_Chairs · 2026-01-26

Accept (Poster)